# Counterfactual Explanations on Robust Perceptual Geodesics

**Eslam Zaher**[1,2]**, Maciej Trzaskowski**[1,3,4]**, Quan Nguyen**[1,3,5]**, Fred Roosta**[1,2]

[1] ARC Training Centre for Information Resilience (CIRES)
[2] School of Mathematics and Physics, University of Queensland
[3] Institute for Molecular Bioscience, University of Queensland
[4] Profenso      [5] QIMR Berghofer Medical Research Institute

## Abstract

Latent-space optimization methods for counterfactual explanations—framed as minimal semantic perturbations that change model predictions—inherit the ambiguity of Wachter et al.'s objective: the choice of distance metric dictates whether perturbations are meaningful or adversarial. Existing approaches adopt flat or misaligned geometries, leading to off-manifold artifacts, semantic drift, or adversarial collapse. We introduce Perceptual Counterfactual Geodesics (PCG), a method that constructs counterfactuals by tracing geodesics under a perceptually Riemannian metric induced from robust vision features. This geometry aligns with human perception and penalizes brittle directions, enabling smooth, on-manifold, semantically valid transitions. Experiments on three vision datasets show that PCG outperforms baselines and reveals failure modes hidden under standard metrics.

## 1 Introduction

As deep learning models grow in scale and impact, interpretability becomes paramount as it offers a crucial lens into their internal reasoning. Traditional saliency-based methods, which highlight influential input features (Simonyan et al., 2014; Sundararajan et al., 2017; Smilkov et al., 2017; Kapishnikov et al., 2021; Ribeiro et al., 2016; Selvaraju et al., 2016; Lundberg & Lee, 2017), have been widely adopted for vision models but produce static, often noisy attributions that lack guidance on how predictions could be altered. **Counterfactual explanation (CE)** methods have emerged as a complementary paradigm grounded in the fundamental human capacity to contemplate *"what if?"* scenarios (Wachter et al., 2017; Ustun et al., 2019; Joshi et al., 2019; Artelt & Hammer, 2019). Rather than merely highlighting salient regions, CEs specify which semantic features should be modified—and how—to produce a different prediction. Wachter et al. (Wachter et al., 2017) formalized this notion as a solution to an optimization problem:

$$\min_{x} \quad \underbrace{r(x^{\star}, x)}_{\text{Similarity Distance}} + \lambda \underbrace{\ell(f(x), y')}_{\text{Classification Loss}} , \tag{1}$$

where $x^{\star}$ is the original input, $y'$ the desired class, $f$ the classifier, $\ell$ a loss function (e.g., cross-entropy), $r$ a distance metric, and $\lambda$ a hyperparameter balancing classification and similarity.

Considerable debate has emerged around whether a CE is fundamentally distinct from an adversarial example (AE), as both arise from the same optimization problem (Wachter et al., 2017; Browne & Swift, 2020; Pawelczyk et al., 2022; Freiesleben, 2022). The choice of distance metric $r$ plays a central role: while it may support meaningful CEs, it can also encourage AEs if it favors imperceptibly small, distributed perturbations. Wachter et al. Wachter et al. (2017) acknowledged this ambiguity, noting that "AEs are counterfactuals by another name," proposing distinction on two grounds: (i) a misalignment of the distance metric with meaningful feature changes—since metrics typically used for AEs favor such dispersed modifications, thereby diminishing their explanatory value, and (ii) adversarial perturbations are non-semantic signals that displace inputs out of the possible world—i.e., off-manifold regions that do not correspond to valid examples under the data distribution.

Rather than directly solving eq. (1), some approaches leverage generative models to produce visual CEs by exploiting low-dimensional semantic representations (Augustin et al., 2022; Mertes et al.,

2022; Looveren et al., 2021; Singla et al., 2020; Lang et al., 2021; Khorram & Fuxin, 2022). For instance, Singla et al. (2020) trained a conditional GAN to produce exaggerated CEs, while Lang et al. (2021) used a conditional STYLEGAN2-based approach to generate sparse visual CEs along disentangled classifier-relevant style-space directions. Khorram & Fuxin (2022) used cycle-consistent losses to train transformations between factual and counterfactual distributions in generative latent spaces. Though visually compelling, these methods rely on exhaustive techniques that depart from the direct optimization formulation and ignore the geometry of the data manifold.

Other research adopt eq. (1) in the latent space of generative models (Joshi et al., 2019; Duong et al., 2023; Dombrowski et al., 2024; Pegios et al., 2024), but either assume flat Euclidean geometry (Joshi et al., 2019; Dombrowski et al., 2024), failing to capture the manifold's intrinsic curvature, or use geometrically informed yet adversarially vulnerable distance metrics (Pegios et al., 2024). For example, REVISE (Joshi et al., 2019) solves the objective in eq. (1) in a VAE latent space under Euclidean assumptions, using explicit $\ell_1/\ell_2$ distance terms. Dombrowski et al. (2024) discard explicit similarity terms and employ Stochastic Gradient Descent (SGD) assuming flat geometry misaligned with the underlying data manifold.

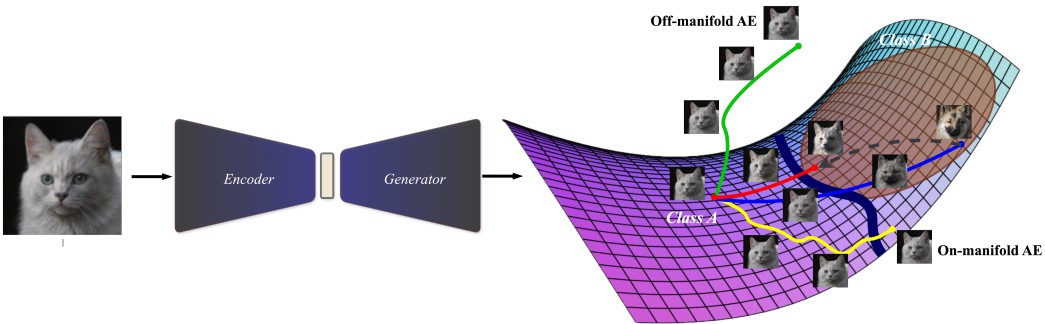

Figure 1: **Schematic of PCG.** An input is mapped through an encoder-generator pair. A linear latent path to a perceptually plausible target-class sample (Class B, brown region) is refined in Phase 1 into the blue geodesic by minimizing robust perceptual energy. In Phase 2, the endpoint and intermediate points are jointly optimized under classification loss and robust energy, resulting in the red counterfactual geodesic. The green trajectory (REVISE, VSGD) ignores manifold geometry, strays off-manifold and produces off-manifold AEs. The yellow trajectory (RSGD/-C) conforms to a fragile geometry, getting stuck in on-manifold adversarial regions (Class B, outside brown region).

This misalignment often causes perturbations to stray off-manifold, leading to implausible or off-manifold AEs. Pegios et al. (2024) proposed equipping the latent space with a Riemannian metric induced by the generator and optimizing with Riemannian SGD (RSGD) to account for the geometry of the data manifold. However, their induced metric is typically derived by pulling back either the pixel-space $\ell_2$ or a standard classifier's feature space metric. Both are problematic in the vision domain: the $\ell_2$ norm is a poor proxy for human perception (Sinha & Russell, 2011; Jordan et al., 2019; Rybkin, 2022), while a standard feature-based metric is semantically brittle as it inherits the adversarial vulnerabilities of non-robust vision models (Sjögren et al., 2022; Ghazanfari et al., 2024b).

Such methods acutely fail in the high-dimensional vision domain, where the counterfactual optimization process can't distinguish between CEs and AEs. Browne & Swift (2020) proposed the notion of a *semantic divide*—a distinction between perturbations that affect human-understandable semantic features or low-level, uninterpretable features. Perturbations with rich semantic content fall on the explanatory side; pixel-level or low-level ones fall on the adversarial side. Browne & Swift (2020) argue that neither distance metrics nor appeals to "possible worlds" fully resolve this distinction; instead, semantic relevance only determines whether a result is a valid CE or an AE.

We agree with Browne & Swift (2020) that the second criterion proposed by Wachter et al.– displacement to off-manifold regions–fails to adequately differentiate AEs from CEs. Several studies have shown that on-manifold AEs exist (Ilyas et al., 2019; Garcia et al., 2023; Song et al., 2018), and can be generated via generative models (Stutz et al., 2019; Zhao et al., 2018), representing a subclass of AEs that reside within Wachter et al.'s "possible worlds". However, we challenge the assertion that distance metrics are inherently incapable of making the distinction. We show that if the

data manifold is endowed with a semantically robust Riemannian metric, solving the counterfactual optimization–when guided appropriately–can cross the semantic divide and produce valid CEs.

**Failure Modes of Previous Approaches.** We attribute the failure of previous latent-space counterfactual optimization methods in the high-dimensional image data regime to three core limitations:

    **(i) Off-manifold Traversal.** Optimization in latent space often disregards the geometry of the data manifold, leading to off-manifold AEs or semantically implausible counterfactuals (Pegios et al., 2024).

    **(ii) Local Gradient Optimization.** Without global structural guidance, single-point geometry-aware gradient methods operate locally and overlook the global manifold structure, including the existence of on-manifold adversarial regions. As a result, they often converge to either semantically distant counterfactuals or on-manifold AEs.

    **(iii) Versatility of Generators.** Even when accounting for manifold geometry, high-capacity generators can exploit non-robust or misaligned distance metrics to produce on-manifold AEs (Stutz et al., 2019; Zhao et al., 2018; Gilmer et al., 2018), fooling the metric rather than producing semantically meaningful perturbations that genuinely cross the semantic barrier.

**Contributions.** Motivated by findings in adversarial robustness that show robust models exhibit perceptually aligned gradients (Ganz et al., 2023b; Srinivas et al.; Shah et al., 2021; Kaur et al., 2019b), robust saliency maps (Etmann et al., 2019; Zhang & Zhu, 2019; Tsipras et al., 2019), and meaningful CEs (Boreiko et al., 2022; Santurkar et al., 2019; Augustin et al., 2020), we introduce a semantically grounded, data-manifold-based approach for perceptually progressive CEs. We emphasize that our focus lies not in interpreting robust classifiers themselves, but in generating explanations for standard models, positioning our work orthogonally to efforts aimed at explaining robust models (Boreiko et al., 2022; Santurkar et al., 2019; Augustin et al., 2020). Our key contributions are as follows:

    **(i) Counterfactual Generation:** We introduce **Perceptual Counterfactual Geodesics (PCG)**, which leverages a robust Riemannian metric on the latent space of a STYLEGAN2/3 generator (Karras et al., 2020b; 2021). This metric is induced from feature spaces of robust vision models. PCG optimizes counterfactual trajectories along geodesic paths, ensuring that counterfactual evolution adheres to robust perceptual perturbations that cross the semantic barrier, avoiding off- or on-manifold adversarial regions.

    **(ii) Perceptual Geodesic Interpolation:** We show that the robust latent geometry underlying PCG enables smooth and semantically robust interpolations between samples. Our experiments demonstrate that trajectories aligned with the robust Riemannian metric preserve class coherence and perceptual structure. In contrast, other metrics collapse into visually ambiguous or brittle transitions due to geometric misalignment.

## 2 BACKGROUND

### 2.1 DIFFERENTIAL GEOMETRY OF DEEP GENERATIVE MODELS

Deep generative models, such as VAEs and GANs, offer a powerful framework for learning high-dimensional data distributions through low-dimensional latent representations (Kingma & Welling, 2022; Higgins et al., 2016; Goodfellow et al., 2014; Karras et al., 2018). These models define a generative function $g : Z \to X$, where $Z \subset \mathbb{R}^d$ is a latent space and $X \subset \mathbb{R}^D$ is a high-dimensional data space, typically $d \ll D$. The image of $Z$ under $g$, denoted $\mathcal{M} = g(Z) \subset X$, forms a subset of the data space, often referred to as the *data manifold*. Under mild regularity conditions—such as smoothness of $g$ with a full-rank Jacobian mapping $J_g \triangleq \partial g / \partial z : Z \to \mathbb{R}^{D \times d}$—this image is a smooth, $d$-dimensional immersed submanifold of $X$ (Shao et al., 2017; Arvanitidis et al., 2017). This construction supports the manifold hypothesis, which posits that real-world high-dimensional data concentrates near such a low-dimensional manifold (Brahma et al., 2016; Fefferman et al., 2013; Tenenbaum et al., 2000).

However, while $Z$ is typically treated as Euclidean, this assumption misaligns with the geometry induced by $g$, as the nonlinear generator significantly distorts its structure. As a result, distances and directions in $Z$ do not reflect the true relationships of the data manifold. This motivates equipping the latent space with a geometry that faithfully reflects the structure of the image manifold $\mathcal{M}$.

## 2.2 PULLBACK METRICS AND THE GEOMETRY OF GENERATORS

A smooth manifold $\mathcal{M} \subset X$ inherits a tangent space $T_x\mathcal{M}$ at each point $x \in \mathcal{M}$, consisting of directions along which one can move locally. To measure lengths and angles, we define a smoothly varying inner product $\langle \cdot, \cdot \rangle_x$ on each tangent space. This defines a Riemannian metric $G(x)$, and the pair $(\mathcal{M}, G)$ forms a Riemannian manifold.

Given a smooth generator $g : Z \to X$, we equip the latent space $Z$ with a Riemannian metric via pullback from the ambient space $X$, assumed to have a metric $G_X(x) \in \mathbb{R}^{D \times D}$. For any $u, v \in T_z Z \cong \mathbb{R}^d$, we define:

$$\langle u, v \rangle_z := \langle J_g(z)u, J_g(z)v \rangle_{G_X(g(z))} = u^\top J_g(z)^\top G_X(g(z)) J_g(z)v,$$

where $J_g(z)$ is the Jacobian of $g$ at $z$. If $J_g(z)$ has full column rank, this defines the pullback metric as $G_Z(z) = J_g(z)^\top G_X(g(z)) J_g(z)$.

While mathematically well-defined, this construction inherits the limitations of the ambient metric. When $G_X(x) = I$, the geometry is induced from the canonical pixel-wise $\ell_2$ metric. In high-dimensional vision tasks, such distances misalign with human perception and are highly sensitive to small, imperceptible perturbations. This issue is not limited to Euclidean metrics; it also applies to other ambient geometries that lack robust semantic grounding. For example, Pegios et al. (2024) pulls back a feature-based metric from a standard classifier, which operates in feature space but still inherits the adversarial vulnerabilities of non-robust models. As a result, the induced latent geometry reflects local structure relative to a brittle and semantically misaligned notion of similarity, often leading to adversarial trajectories (Browne & Swift, 2020).

## 2.3 LATENT SPACE COUNTERFACTUAL OPTIMIZATION

We summarize several methods that solve variations of eq. (1) in the latent space of generative models.

**REVISE.** Joshi et al. (2019) introduced an approach based on VAEs for tabular data, where the latent code $z$ of an input $x^\star$ is updated via SGD on the objective $\mathcal{L} = d(x^\star, g(z)) + \lambda \ell(f(g(z)), y')$. This method relies on two assumptions: that pixel-wise Euclidean distances in ambient space provide meaningful similarity, and that Euclidean SGD updates in latent space correspond to smooth semantic transitions. Both assumptions fail in high-dimensional vision domains, where distances are misaligned with perception and SGD updates stray off-manifold.

**Vanilla SGD (VSGD).** To adapt to vision settings, Dombrowski et al. (2024) proposed eliminating the distance term in REVISE and directly applying vanilla SGD to the classification loss:

$$z \leftarrow z - \eta \nabla_z \Big[ \ell \big( f(g(z)), y' \big) \Big].$$

While sidestepping metric misalignment in $X$, it still assumes a flat Euclidean geometry in $Z$, ignoring the curvature induced by $g$. Since $g$ is highly nonlinear in expressive models, such updates often stray off the manifold and lead to off-manifold AEs or perceptually implausible counterfactuals.

**Riemannian SGD (RSGD).** Pegios et al. (2024) proposed RSGD to account for the curvature of the data manifold by replacing Euclidean gradients with Riemannian ones derived from a pullback metric on the latent space. Given a stochastic VAE generator $g_\varepsilon(z) = \mu(z) + \sigma(z) \odot \varepsilon$, with $\varepsilon \sim \mathcal{N}(0, I)$, the latent metric is defined as the expected pullback of the ambient $\ell_2$ metric:

$$\hat{G}_Z(z) \approx J_\mu(z)^\top J_\mu(z) + J_\sigma(z)^\top J_\sigma(z),$$

and optimization proceeds via: $\quad z \leftarrow z - \eta \dfrac{r}{\|r\|_2}, \quad$ where $\quad r = \hat{G}_Z(z)^{-1} \nabla_z \ell(f(\mathbb{E}_\varepsilon[g_\varepsilon(z)]), y')$.

A variant, RSGD-C, replaces the ambient metric with the pullback of a classifier-based feature metric, using the final-layer representation of a standard classifier. This introduces task-awareness by aligning updates with decision-relevant directions.

Both methods remain limited by their underlying metrics. Pixel-wise $\ell_2$ distances are fragile and misaligned with perception, and standard classifier-based features inherit adversarial vulnerabilities. RSGD/-C does not enforce geodesic paths and has been applied only in low-dimensional domains where adversariality is less evident.

## 3 METHODOLOGY

Prior approaches fail in the vision domain due to three tightly coupled issues: the use of perceptually misaligned metrics (e.g., $\ell_2$ in pixel space or fragile classifier-based metrics), reliance on local gradient updates that ignore global manifold structure, and the expressive power of high-capacity generators that exploit these misalignments to produce adversarial perturbations.

Our method, **PCG**, addresses these limitations by casting counterfactual generation as a global curvature-aware optimization over latent trajectories on a Riemannian manifold, where the generator induces a latent geometry aligned with human perception. To define this geometry, we construct a perceptually robust ambient metric. Unlike standard classifiers, robust models learn representations that are resistant to adversarial perturbations and aligned with human perceptual similarity. These robust intermediate activation spaces exhibit linearly separable structure and encode grounded, semantically meaningful features. As a result, the Euclidean metric becomes a more reliable proxy for perceptual similarity in these robust semantic spaces, unlike its failure in pixel or fragile semantic spaces. We leverage this structure to define a composite ambient metric by aggregating pullbacks of the Euclidean metric from robust feature spaces into the input space, capturing hierarchical, perceptually coherent variations. Formally, we define the robust perceptual metric as:

$$
G_R(x) = \sum_{k=1}^{K} w_k \, J_{h_k}(x)^\top J_{h_k}(x), \quad w_k = \frac{1}{N_k},
$$

where $K$ is the number of selected intermediate layers of a pretrained robust vision model, $h_k(x)$ denotes the activation of the $k$-th layer with dimensionality $d_k \ll D$, $J_{h_k}(x) \in \mathbb{R}^{d_k \times D}$ is its Jacobian with respect to the input $x \in \mathbb{R}^D$, and $N_k$ denotes the total size (number of elements) of the activation $h_k(x)$, which normalizes each layer so that no single feature space dominates due to its size. Pulling back $G_R$ through the generator $g : Z \to X$ defines the latent-space metric

$$
G_Z(z) = J_g(z)^\top G_R(g(z)) J_g(z),
$$

which induces a latent geometry that penalizes brittle or non-robust directions and favors perturbations that produce perceptually smooth, semantically aligned variations in the image space.

We seek a smooth latent trajectory $\gamma : [0, 1] \to Z$ such that $g(\gamma(t))$ evolves through robust semantic regions. The perceptual length of this trajectory, where $\gamma'(t) = d\gamma/dt$ is the latent-space velocity, evaluated under $G_R$, is

$$
L(g(\gamma)) = \int_0^1 \sqrt{\gamma'(t)^\top G_Z(\gamma(t))\gamma'(t)}dt,
$$

and minimizing this length under constant-speed parametrization is equivalent to minimizing the robust perceptual energy (Jost, 2017):

$$
E(g(\gamma)) = \frac{1}{2} \int_0^1 \gamma'(t)^\top G_Z(\gamma(t))\gamma'(t)dt. \tag{2}
$$

Expanding $G_Z$ using the composite metric shows that the pullback energy is a weighted sum of squared velocities in each robust feature space:

$$
\gamma'(t)^\top G_Z(\gamma(t))\gamma'(t) = \sum_{k=1}^{K} w_k \left\| \frac{d}{dt} h_k\big(g(\gamma(t))\big) \right\|_2^2.
$$

Minimizing $E(g(\gamma))$ thus amounts to finding a geodesic whose generator outputs move smoothly and consistently across all robust semantic layers. To do this, we discretize $\gamma$ into $T + 1$ points $\{z_0, \ldots, z_T\}$, where $z_0$ is the latent encoding of the input $x^\star$, and $z_T$ is initialized as the latent encoding of an arbitrary target-class sample from the dataset. This initialization is critical: unlike previous methods that perform iterative updates from a single starting point—which often converge to on-manifold adversarial endpoints—we initialize between two manifold-conforming points to guide global transitions across semantically valid regions under the robust metric. Using forward finite differences as in Shao et al. (2017), we approximate the robust feature-space velocity at $t_i$

as $dh_k(g(\gamma(t)))/dt \mid_{t=t_i} \approx (h_k(g(z_{i+1})) - h_k(g(z_i)))/\delta t$. This gives the discrete robust energy equivalent of eq. (2):

$$E_{\text{robust}}(\mathbf{z}) = \frac{1}{2}\sum_{i=0}^{T-1}\sum_{k=1}^{K}\frac{w_k}{\delta t}\left\|h_k\big(g(z_{i+1})\big) - h_k\big(g(z_i)\big)\right\|_2^2, \quad \text{where } \mathbf{z} \triangleq [z_0, \ldots, z_T] \text{ and } \delta t = 1/T.$$

Optimization proceeds in two stages. In Phase 1, we fix $z_0$ and $z_T$ and minimize $E_{\text{robust}}(\mathbf{z})$ with respect to the intermediate points to obtain a geodesic consistent with the robust semantic geometry induced by the generator. In Phase 2, we release $z_T$ and jointly optimize the energy and a classification loss to ensure the endpoint maintains the desired prediction under $f$. The combined loss is

$$\mathcal{L}(\mathbf{z}) = E_{\text{robust}}(\mathbf{z}) + \lambda \cdot \ell\big(f(g(z_T)), y'\big).$$

In practice, Phase 2 is implemented as a coarse-to-fine refinement of the Phase 1 geodesic. We start from the path connecting $x_{\text{orig}}$ to a target-class exemplar and optimize $\mathcal{L}$ for a fixed budget, with a small initial $\lambda$ that is increased over time. This schedule gradually shifts the optimization from purely geometric regularization (when $\lambda$ is small and $E_{\text{robust}}$ dominates) towards enforcing the target-class prediction at the endpoint (as $\lambda$ grows). To avoid either collapsing back to the input or drifting too far into the target region, we periodically apply a re-anchoring step: at fixed intervals, we scan along the current path for points that are already classified as the target class and select the one closest to $x_{\text{orig}}$ (in the induced perceptual geometry) as the new endpoint. We then reparameterize the path by inserting midpoints between successive latent codes to restore the original number of waypoints and resume optimization from this shortened, re-anchored trajectory. Iterating this procedure has the effect of progressively "pulling" the endpoint towards the input while keeping the target label fixed, so that the path converges to a counterfactual that remains on the robust geodesic manifold and is as close as possible to $x_{\text{orig}}$ under the induced metric. The overall structure of our two-stage optimization and the contrast with prior methods is illustrated in Figure 1; full algorithm, induced metric, and optimization details are provided in Appendix A.1.

## 4 Experiments

We evaluate PCG against prior latent-space optimization methods. In section 4.1, we first show the failure mode of interpolation methods inherent in their geometrical assumptions, and demonstrate the effect of our proposed robust Riemannian metric in generating perceptually smooth geodesics that underpins PCG. In section 4.2, we compare PCG with other approaches in terms of the perceptual plausibility of the generated counterfactuals. Finally, we quantitatively evaluate PCG under both typical and geometry-aware distance measures. Code for our experiments is available here.

**Datasets.** We evaluate our method on three high-dimensional real-image datasets: (1) AFHQ (Choi et al., 2020), with high-resolution images of cats, dogs, and wild animals; (2) FFHQ (Karras et al., 2019), containing 70,000 diverse human face images; and (3) PlantVillage (Hughes & Salathé, 2015), with labeled images of healthy and diseased plant leaves across species.

**Models.** We train STYLEGAN2 generators from scratch on AFHQ and PlantVillage ($\approx$140 NVIDIA H100 GPU-hours per model) (Karras et al., 2020a). For AFHQ, we also use a pretrained STYLEGAN3 generator (Karras et al., 2021). For FFHQ, we use pretrained STYLEGAN2 and STYLEGAN3. Post hoc, we train image-to-latent encoders (used for all counterfactual optimization in z-space) and then briefly fine-tune the encoder–generator pair jointly. For classifiers, we train binary models based on the VGG-19 backbone (Simonyan & Zisserman, 2014): one per AFHQ class pair and a healthy–vs–unhealthy classifier for PlantVillage. Because FFHQ lacks labels, we train attribute classifiers on CelebA (Liu et al., 2015) and apply them to FFHQ. Architectural and training details appear in Appendix A.3.

**Baselines.** We compare PCG against the following latent-space based approaches:
- **REVISE** (Joshi et al., 2019). Latent-space equivalent of Wachter et al.'s objective based on SGD.
- **VSGD** (Dombrowski et al., 2024). It performs distance-free vanilla SGD in the latent space.
- **RSGD/-C** (Pegios et al., 2024). In these variants, a Riemannian metric is used to guide SGD. The metrics are pull-back from either the Euclidean metric in the ambient space or in the final layer of the classifier under explanation.

## 4.1 Effect of Latent Geometry on Interpolation

In Figure 2, we illustrate how latent-space geometry shapes interpolation. The top row linearly interpolates in latent space Z under a Euclidean assumption, which ignores the nonlinear distortion induced by the generator and produces mid-path off-manifold artifacts such as class ambiguity, unnatural warping, and deformed textures. The second row minimizes pixel-space MSE in X, which induces a latent-space geometry by pulling back the Euclidean metric from X to Z; transitions remain brittle and semantically incoherent, with midway blends of disparate attributes that expose the fragility and misalignment of pixel-wise distances. The third row uses the pullback of a feature metric from a standard ResNet-50 (He et al., 2016) (see appendix A.4); semantics improve, yet fading, illumination shifts, and class discontinuities persist. These instabilities reflect the vulnerability of non-robust models to adversarial perturbations and reliance on brittle features, with similar failure modes reported in Laine (2018) using VGG-19. In contrast, the fourth row applies our robust perceptual metric derived from a robust ResNet-50, producing smooth, on-manifold trajectories with consistent semantics and coherent evolution. This confirms our hypothesis that robust Riemannian geometry enables smooth, semantically valid on-manifold interpolations while avoiding adversarial collapse.

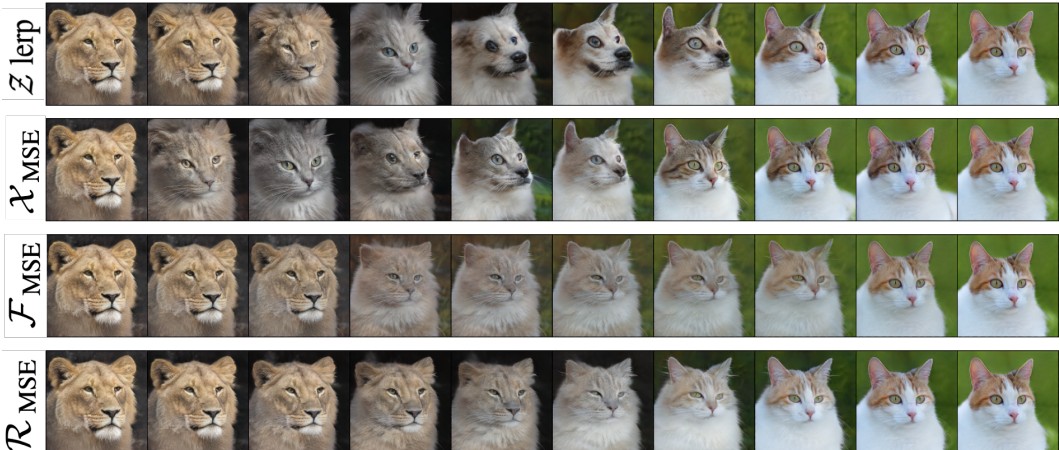

Figure 2: Interpolation paths under four latent geometries based on StyleGAN2 (top→bottom). (a) $Z$-linear (Euclidean): flat latent metric; off-manifold artifacts. (b) Pixel MSE pullback: Euclidean metric pulled back to $Z$; brittle, incoherent paths. (c) Standard feature pullback: non-robust ResNet-50; better semantics but still fading and discontinuities. (d) Robust perceptual pullback (ours): robust ResNet-50; smooth, consistent, on-manifold trajectories. See Appendix B.1 for StyleGAN3 results.

## 4.2 Perceptual Counterfactual Geodesics

Having established smooth perceptual geodesics under our proposed metric, we now demonstrate their refinement into plausible CEs. Figure 3 showcases the two-stage nature of our approach. In Phase 1 (rows 1 and 3), we generate an initial perceptual geodesic between the input and an arbitrary target-class sample, such as a dog image for a cat input, or a non-blonde face for a blonde input. Although the target is semantically distant, the path remains coherent, illustrating the alignment of our metric with perceptual structure. In Phase 2 (rows 2 and 4), we release the endpoint and jointly optimize it with the path under the classification loss, allowing the counterfactual to move closer to the input while maintaining geodesicity. The resulting counterfactual geodesics trace robust regions of the data manifold and maintain consistent semantics throughout the trajectory, retaining the semantic continuity and avoiding adversarial shortcuts or abrupt transitions. This step ensures the whole path travels through perceptually robust regions on the manifold as shown in Fig 1. We show that different choices of the target-class exemplar lead optimization to converge within a small neighborhood of the input, producing diverse yet faithful counterfactual explanations; see Appendix B.3

**Comparison with Baselines.** We now evaluate the final counterfactuals produced by PCG against existing latent-space optimization methods. As shown in Figure 4, our method consistently produces semantically valid CEs that remain close to the input while effecting the desired class transition. In

contrast, RSGD and RSGD-C, despite accounting for local curvature, rely on fragile metrics (e.g., pixel-space $\ell_2$ or non-robust classifier features) that remain vulnerable to adversarial manipulation. Many of the generated counterfactuals collapse into on-manifold AEs—as seen in rows 1, 2, 4, 5, and 6. Like the outputs of other baselines, they fall on the adversarial side of the semantic divide. Even when RSGD variants converge (e.g., row 3), the output is visibly distant from the input in pose and structure, reflecting the lack of geodesic constraint and a tendency to traverse longer manifold paths. VSGD, which assumes flat Euclidean geometry, produces off-manifold perturbations that are either perceptually implausible, or adversarial. In row 2, the generated counterfactual exhibits class ambiguity and disoriented eye alignment; in row 3, the face is unnaturally elongated with distortions under the chin; in row 6, the leaf counterfactual contains an unnatural cusp-like protrusion that breaks the expected symmetry, fullness, and surface continuity of leaves. These artifacts arise from ignoring the data manifold altogether. REVISE exhibits similar failure modes: the strong pixel-wise distance penalty constrains outputs to remain close in $\ell_2$ norm, but adversarial. All REVISE outputs in the figure represent off-manifold AEs, driven by the optimization pressure to minimize distance rather than induce meaningful semantic change. In contrast, PCG navigates robust regions of the manifold along perceptual geodesics, producing minimal, semantically faithful changes.

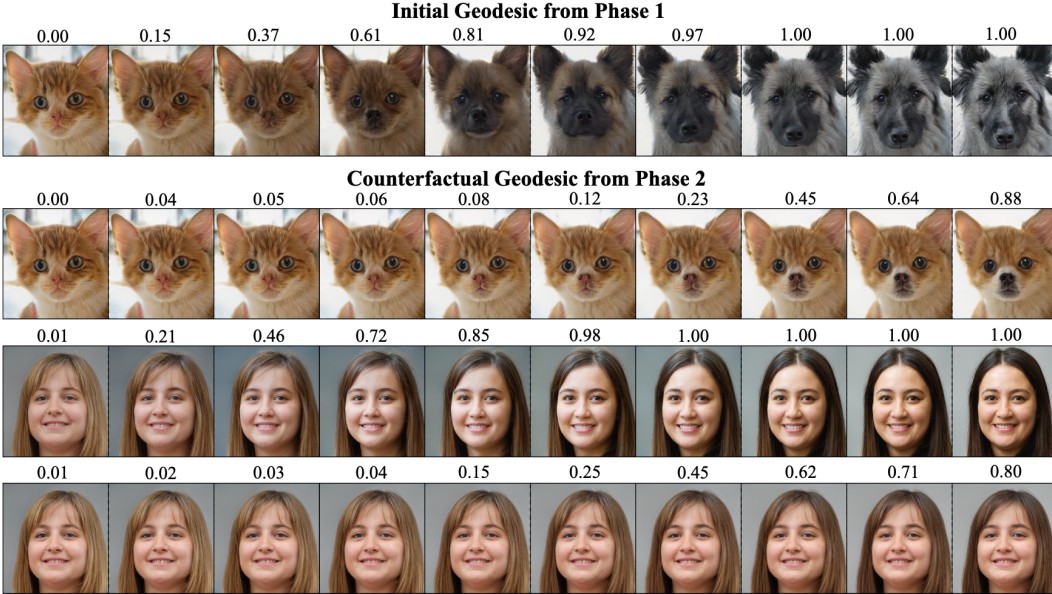

Figure 3: Perceptual Counterfactual Geodesics. Rows 1 and 3: initial geodesics from Phase 1 between an input and a target-class sample. Rows 2 and 4: counterfactual geodesics after Phase 2, where the endpoint is optimized with the path. from Phase 2 stay in robust regions of the manifold and preserve semantic continuity. Results from STYLEGAN2 (see Appendix B.2 for STYLEGAN3)

Table 1: Quantitative comparison across datasets for STYLEGAN2 (see Appendix B.4 for STYLE-GAN3 and Appendix B.5 for runtime complexity). Columns report $\mathcal{L}_1$ (pixel $\ell_1$), $\mathcal{L}_2$ (pixel $\ell_2$), $\mathcal{L}_\mathcal{F}$ (pullback from standard VGG-16), and $\mathcal{L}_\mathcal{R}$ (pullback from robust Inception-V3). Lower is better.

| Method | AFHQ | | | | FFHQ | | | | PlantVillage | | | |
|---|---|---|---|---|---|---|---|---|---|---|---|---|
| | $\mathcal{L}_1$ | $\mathcal{L}_2$ | $\mathcal{L}_\mathcal{F}$ | $\mathcal{L}_\mathcal{R}$ | $\mathcal{L}_1$ | $\mathcal{L}_2$ | $\mathcal{L}_\mathcal{F}$ | $\mathcal{L}_\mathcal{R}$ | $\mathcal{L}_1$ | $\mathcal{L}_2$ | $\mathcal{L}_\mathcal{F}$ | $\mathcal{L}_\mathcal{R}$ |
| REVISE | 1.20±0.12 | **0.73**±0.18 | 1.08±0.10 | 2.70±0.05 | 0.82±0.08 | **0.32**±0.13 | 0.82±0.08 | 2.78±0.06 | 0.50±0.13 | **0.38**±0.15 | 0.96±0.06 | 2.87±0.07 |
| VSGD | 1.31±0.11 | 1.49±0.15 | 1.60±0.09 | 2.90±0.08 | 0.79±0.11 | 0.96±0.10 | 1.50±0.12 | 2.86±0.07 | 0.83±0.13 | 0.94±0.17 | 1.18±0.07 | 3.01±0.09 |
| RSGD | 0.85±0.08 | 1.32±0.09 | 0.70±0.07 | 1.85±0.05 | 0.61±0.05 | 0.84±0.07 | 0.61±0.04 | 2.41±0.05 | 0.78±0.08 | 0.82±0.11 | 0.54±0.05 | 2.28±0.04 |
| RSGD-C | 0.93±0.10 | 1.45±0.17 | 0.65±0.08 | 1.75±0.06 | 0.68±0.06 | 0.93±0.09 | 0.48±0.04 | 2.11±0.04 | 0.80±0.10 | 0.86±0.13 | 0.45±0.05 | 2.03±0.06 |
| PCG (ours) | **0.79**±0.07 | 1.14±0.10 | **0.53**±0.06 | **0.31**±0.02 | **0.42**±0.03 | 0.72±0.09 | **0.39**±0.05 | **0.22**±0.06 | **0.36**±0.03 | 0.56±0.05 | **0.34**±0.04 | **0.20**±0.05 |

**Distance-based Evaluation.** We assess counterfactual proximity using four distance metrics: $\mathcal{L}_1$ (pixel-wise $\ell_1$), $\mathcal{L}_2$ (pixel-wise $\ell_2$), $\mathcal{L}_\mathcal{F}$ (distance induced by the pullback from standard ResNet-50 features), and $\mathcal{L}_\mathcal{R}$ (pullback from robust ResNet-50 features). Each induced metric is computed between the input and the final counterfactual in image space using the local quadratic form $\mathcal{L}_G(z_0, z_T) = \sqrt{(g(z_T) - g(z_0))^\top G(g(z_0))(g(z_T) - g(z_0))}$, where $G \in \{G_\mathcal{F}, G_\mathcal{R}\}$ is the respec-

tive ambient metric. This approximates perceptual distance in the feature space around the input. To avoid entanglement between optimization and evaluation, we compute $\mathcal{L}_\mathcal{F}$ using an independent VGG-16 model that was never involved in training or counterfactual optimization, and we compute $\mathcal{L}_\mathcal{R}$ using a robustly trained Inception-V3 model (Alfarra et al., 2022b) separate from the robust ResNet-50 that defines our metric. As shown in Table 1, our method achieves the lowest distances across all geometry-aware metrics and also under $\mathcal{L}_1$, indicating sparse, perceptually meaningful changes. The margin is largest under $\mathcal{L}_\mathcal{R}$, and extends to $\mathcal{L}_\mathcal{F}$, since our robust geodesics stay closer even under weaker perceptual proxies. REVISE and VSGD often stray off-manifold, producing AEs that appear close under $\mathcal{L}_2$ (unsurprisingly, as REVISE directly minimizes this metric) but deviate sharply in all perceptual geometries. RSGD and RSGD-C operate under their metrics, but lack geodescity and remain vulnerable to on-manifold AEs—perturbations smooth under $\ell_2$ and $\mathcal{L}_\mathcal{F}$ yet semantically fragile. These cases highlight that our proposed $\mathcal{L}_\mathcal{R}$ serves as a more faithful evaluation metric, exposing failure modes that remain hidden under non-robust distances. Low scores in $\mathcal{L}_1, \mathcal{L}_2$, or $\mathcal{L}_\mathcal{F}$ do not guarantee proximity and can coincide with adversarial behavior.

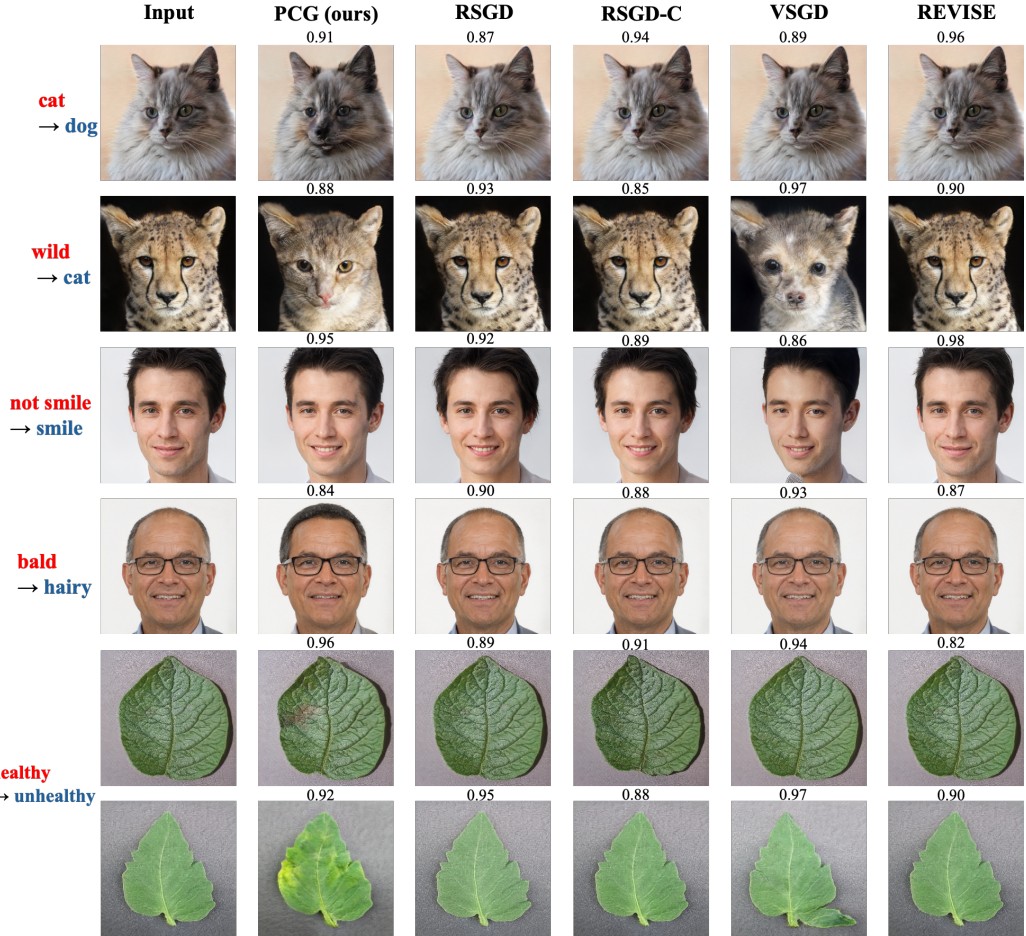

Figure 4: Qualitative comparison of counterfactuals across methods with STYLEGAN2. Columns show input images followed by counterfactuals from PCG (ours), RSGD, RSGD-C, VSGD, and REVISE. Rows indicate input and target /class. PCG produces minimal, semantically faithful changes along robust geodesics, while baselines often show off-manifold artifacts, semantic drift, or adversarial collapse. Optimization details for baselines are presented in Appendix A.2.

**Explanandum-based Metrics.** We quantify realism using the standard FID (Heusel et al., 2018) and its robust variant R-FID (Alfarra et al., 2022a). As a sparsity measure, we report a representation-based COUT following Khorram & Fuxin (2022). For closeness to the original input, we use LPIPS (Zhang et al., 2018) and its robust counterpart R-LPIPS (Ghazanfari et al., 2023). For validity, we

propose a Semantic Margin (SM) metric to evaluate whether generated counterfactuals move into regions of the data space that are genuinely associated with the target class. We further use a Manifold Alignment Score (MAS) to measure how the direction of change between original images and their counterfactuals aligns with manifolds induced by different geometries. Finally, we report flip rate for completeness. All evaluation metrics are detailed in Appendix A.5. We additionally evaluate the smoothness of our counterfactual geodesics (Appendix B.7), scaling of our method with respect to image resolution and path length (Appendix B.5 ), ablations on $\lambda$ (Appendix B.6), the effect of the selected robust backbone and layer aggregation scheme (Appendix B.8), and more results for classifiers beyond the VGG-19 backbone extended to multiclass classification, and comparisons with counterfactuals from robust models (Appendix B.9).

Table 2 summarizes the explanandum-based metrics discussed above. Under standard FID, PCG attains the lowest score, but standard FID alone is known to be insensitive to adversarial artifacts (Alfarra et al., 2022a). When we move to the robust variant R-FID, the gap widens: PCG remains close to the real target distribution, whereas baselines degrade more strongly, indicating that their improvements in standard FID are at least partly driven by non-robust directions. LPIPS and its robust counterpart R-LPIPS quantify the perceptual displacement between original images and their counterfactuals; PCG achieves the smallest distances, with the clearest separation under R-LPIPS, in line with the fact that our trajectories are constructed as geodesics in a robust perceptual geometry and therefore change content more gradually in robust feature space. The COUT scores follow the same pattern: PCG induces more concentrated, lower-magnitude changes in the classifier's internal representation than baseline methods, rather than diffusing changes across many low-level features. Finally, the mean SM shows that PCG counterfactuals move into regions of robust feature space that are genuinely populated by target-class data (positive margins), while baseline methods more frequently remain in mixed or non-target neighborhoods. Finally, MAS scores reveal how the direction of change interacts with different geometries: methods that do not explicitly model geometry (e.g, REVISE, VSGD) exhibit weak alignment or specialize to the geometry they implicitly optimize (e.g, RSGD), whereas PCG achieves strong alignment in the robust feature space it induces, and this behavior generalizes to standard feature geometry, consistent with its construction as a robust perceptual geodesic. Taken together, these metrics support our central claim that standard FID/LPIPS alone can be overly optimistic in adversarially vulnerable regimes, whereas their robustified counterparts and semantic diagnostics (COUT, mean SM, and MAS) draw out the advantage of semantics-aware, robustly induced geodesics over existing latent-space counterfactual methods much more clearly.

Table 2: Evaluation results across realism, closeness, faithfulness, manifold-alignment metrics, and flip rate for various methods (STYLEGAN2 on AFHQ).

| Method | Realism ↓ | | Closeness ↓ | | Faithfulness ↑ | | MAS ↑ | | | Flip ↑ |
| | FID | R-FID | LPIPS | R-LPIPS | COUT | Mean SM | Pixel | Standard | Robust | Rate |
|---|---|---|---|---|---|---|---|---|---|---|
| REVISE | 18.5 | 50.1 | 0.85 | 0.67 | 0.09 | -0.48 | 0.21 | 0.18 | 0.14 | **98%** |
| VSGD | 23.5 | 46.7 | 0.93 | 0.79 | 0.10 | -0.14 | 0.17 | 0.21 | 0.19 | 92% |
| RSGD | 12.9 | 37.8 | 0.61 | 0.68 | 0.13 | 0.03 | **0.82** | 0.45 | 0.21 | 96% |
| RSGD-C | 12.7 | 28.3 | 0.59 | 0.53 | 0.25 | 0.05 | 0.68 | 0.84 | 0.47 | 94% |
| PCG (ours) | **8.3** | **9.1** | **0.24** | **0.17** | **0.43** | **0.74** | 0.65 | **0.87** | **0.91** | 95% |

## 5 CONCLUSION

We introduced Perceptual Counterfactual Geodesics (PCG), a method for generating semantically faithful counterfactuals by optimizing smooth trajectories on a latent Riemannian manifold equipped with a robust perceptual metric. Our two-phase framework operationalizes established ideas from pullback geometry and robust perception into a practical algorithm. Empirically, PCG outperforms latent-space baselines and avoids their common failure modes (off- and on-manifold adversarial collapse, semantic drift). In addition, the robust geometry-aware evaluation $\mathcal{L}_{\mathcal{R}}$ exposes errors that remain hidden under standard distances, providing a more reliable yardstick for counterfactual quality. Conceptually, the contribution is algorithmic: we show that when the latent space is endowed with a robust, perceptually aligned geometry and optimized globally along paths, counterfactuals become smooth, diverse, and faithful. Final notes on our scope, limitations, and future work are discussed in Appendix C

## ETHICS STATEMENT

All authors have read and will adhere to the ICLR Code of Ethics. Our experiments use publicly available vision datasets under their licenses; no new human-subject data were collected, and we do not perform re-identification or demographic inference. Any released code is intended for research use and will include guidance discouraging harmful or deceptive applications. *LLM usage disclosure:* in line with ICLR policy, we used a large language model only for light copy-editing (grammar, typos, minor phrasing/formatting); it did not contribute to research ideation, analysis, or claims.

## REPRODUCIBILITY STATEMENT

All methodological details, derivations, and hyperparameter settings required to reproduce our experiments are described in the main text (Section 3) and in Appendix A.1, where we also provide pseudocode for our two-stage optimization procedure. Architectural specifications, training protocols for generators, encoders, and classifiers, and additional results (including sensitivity to initialization) are included in Appendices A and B. Anonymized source code implementing PCG and all evaluation metrics is provided in Section 4 to enable full replication of our experiments.

## ACKNOWLEDGMENTS

This research was partially supported by the Australian Research Council through an Industrial Transformation Training Centre for Information Resilience (IC200100022). Quan Nguyen is supported by a NHMRC Investigator Grant (GNT2008928). Maciej Trzaskowski is the Managing Director of Profenso; this work was conducted independently of Profenso.

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

## A    FURTHUR DETAILS ON THE PCG ALGORITHM, BASELINES, MODELS, AND METRICS.

### A.1    PCG OPTIMIZATION

Our objective minimizes the discrete robust perceptual energy of a latent trajectory under the pullback geometry. Because we differentiate the energy itself (squared feature increments along $h_k \circ g$, backprop through $h_k$ and $g$ automatically inserts the Jacobian factors that define the pullback metric. Two implications follow. First, in Phase 1 (energy-only), standard gradient descent already converges to a manifold-conforming geodesic for the path variables, so a Riemannian correction brings no additional benefit. Second, in Phase 2 we add a classification term that touches only the endpoint $z_T$; while one could Riemannian-correct that update in isolation, it is unnecessary in our coupled objective: the energy term continues to regularize all latent points (including $z_T$), steering the entire trajectory to remain a counterfactual geodesic.

PCG proceeds in two phases. The first constructs a smooth geodesic path between the input and a target-class sample, optimized for 200 steps with a fixed learning rate of $1e-3$. The second refines the path into a faithful counterfactual over 300 steps, using the same learning rate and a dynamic $\lambda$ schedule: starting from $1e-4$ and multiplying by 5 every 50 steps. At each such interval, we apply a re-anchoring strategy: the path endpoint is reassigned to the closest point to the input along the trajectory that is classified as belonging to the target class. We then increase the resolution of the path by inserting midpoints between each pair of consecutive latent codes, restoring the original path length. Optimization resumes to refine the updated path, progressively giving closer counterfactuals. For completeness, the PCG optimization pseudocode is given in Algorithm 1.

### A.2    BASELINES OPTIMIZATION

To ensure comparability, all baselines start from the same initialization $z_0 = e(x^\star)$, use the same encoder–generator pair, and are optimized for the same number of steps as PCG $(200 + 300)$. We use Adam (Kingma & Ba, 2015) and select the step size by a small sweep $\eta \in \{1e-4, 3e-4, 1e-3, 1e-2\}$ on a held-out split; we report the best setting per method.

**VSGD.** Vanilla latent descent minimizes only the classification loss $\ell(f(g(z)), y')$ (no similarity term, no $\lambda$). We run Adam with the learning-rate sweep above.

**REVISE.** We optimize $d(x^\star, g(z)) + \lambda \, \ell(f(g(z)), y')$ in latent space. For fairness and due diligence, $\lambda$ follows the same dynamic schedule used in PCG Phase 2 (start $1e-4$, $\times 5$ every 50 steps). We use the same Adam sweep for $\eta$.

**RSGD/-C.** These variants require the inverse of the induced latent metric. We compute the natural-gradient direction by solving $G_Z(z)\, r = \nabla_z \mathcal{L}$ with Conjugate Gradients (Hestenes & Stiefel, 1952), using Jacobian–vector products via autodiff; this avoids explicit Jacobian assembly and matrix inversion. Since the original code targets VAEs on tabular data and is not public, we implement a deterministic metric compatible with our GAN setting (pixel $\ell_2$ pullback for RSGD; classifier-feature pullback for RSGD-C) and apply the same Adam step-size sweep for the outer update.

### A.3    GENERATORS, ENCODERS, AND CLASSIFIERS

**Style-based generator (image prior).** We use the official **StyleGAN2-ADA** (and, where noted, **StyleGAN3**) implementations as our image prior. The generator provides a smooth latent manifold on which we optimize trajectories; we do not introduce architectural modifications beyond standard configuration (resolution/weights).

**Image→latent encoder (inversion).** To place real images on the generator's latent manifold, we train a lightweight encoder that maps an input image to a single latent vector compatible with the generator's input space. Its role is purely representational: enable mapping for endpoints and faithful reconstructions; exact layer choices are not critical to the method.

**Discriminator (training-only).** When (re)training a generator, we use the standard discriminator bundled with the official StyleGAN repositories. It is only a training counterpart—*never* used by our optimization or evaluation procedures.

**Task classifiers (decision function $f$).** For each dataset/attribute, we use a conventional supervised image classifier (e.g., VGG-19 from TorchVision) as the decision function whose prediction we seek to change. These models are straightforward baselines chosen for familiarity and availability; they are not part of the perceptual metric.

**Robust backbones (perceptual geometry & evaluation).** To define our robust perceptual metric and for geometry-aware, we rely on *adversarially trained* ImageNet backbones sourced from public robustness libraries (Engstrom et al., 2019; Debenedetti et al., 2023). These networks are used *only* to induce a perceptually aligned geometry and to score distances; they are distinct from the task classifier $f$.

**Why these choices.** The generator supplies a strong visual prior (manifold parameterization), the encoder puts real data on that manifold, the classifier defines the target decision boundary, and robust backbones define a perceptually grounded geometry. This separation lets us optimize counterfactual *paths* on a high-quality manifold while keeping the decision function and the perceptual metric decoupled.

**Requirements for each method.** Tables 3 and 4 summarize practical requirements and optimization burden. All methods require a generator $g$ and (for real images) an encoder $e$; only PCG additionally uses a robust backbone to induce the perceptual geometry. Unlike RSGD variants, PCG does not perform metric inversion (no CG solves), which keeps its runtime *below* RSGD/RSGD-C despite being path-based; qualitatively it is "Medium," while RSGD and RSGD-C are "High" and "Highest," respectively. REVISE and VSGD remain the lightest due to single-point Euclidean updates without metric operations.

Table 3: Component requirements by method. "Yes/Optional" means the encoder is needed for real-image inversion but optional for synthetic latents.

| Method | Generator $g$ | Encoder $e$ | Classifier $f$ | Robust backbone |
|---|---|---|---|---|
| PCG (ours) | Yes | Yes/Optional | Yes | Yes |
| RSGD-C | Yes | Yes/Optional | Yes | No |
| RSGD | Yes | Yes/Optional | Yes | No |
| REVISE | Yes | Yes/Optional | Yes | No |
| VSGD | Yes | Yes/Optional | Yes | No |

Table 4: Optimization and compute summary. "Metric inversion" refers to solving $G_Z(z)\,r = \nabla_z \mathcal{L}$ (e.g., via Conjugate Gradients).

| Method | Optimization Style | Metric inversion | Relative compute |
|---|---|---|---|
| PCG (ours) | Path optimization (two-phase: energy then energy+cls) | No | Medium |
| RSGD-C | Single-point Riemannian descent (feature-space pullback) | Yes (CG) | Highest |
| RSGD | Single-point Riemannian descent (pixel-space pullback) | Yes (CG) | High |
| REVISE | Single-point Euclidean descent (distance + cls) | NA | Low |
| VSGD | Single-point Euclidean descent (cls only) | NA | Lowest |

### A.4 METRIC COMPOSITION, ROBUST BACKBONES, AND SMOOTHNESS

**Backbone choice.** We instantiate the perceptual geometry using *adversarially trained* ImageNet backbones that supply perceptually aligned, manifold-conforming gradients (Ganz et al., 2023b; Zhang & Zhu, 2019; Tsipras et al., 2019; Ilyas et al., 2019; Stutz et al., 2019). Our default induced geometry in the main text uses an $L_2$-robust ResNet-50 trained on ImageNet with $\varepsilon = 3.0$ (Engstrom et al., 2019), and in the appendix we report analogous results using an $L_2$-robust XCiT-S12 vision transformer trained under the same threat model (Debenedetti et al., 2023). These networks are used only to induce the metric; they are *never* the same model as the task classifier $f$, and their weights remain frozen throughout. The choice of robust backbones is motivated by concrete theoretical and

empirical results in adversarial robustness and Perceptually Aligned Gradients (PAGs). Etmann et al. and follow-up work show that adversarially trained models produce saliency maps that are more strongly aligned with human salient structure and suppress high-frequency, non-salient directions (Etmann et al., 2019; Zhang & Zhu, 2019; Tsipras et al., 2019; Ilyas et al., 2019). Srinivas et al. provide a theoretical account via off-manifold robustness, showing that when a classifier is trained to be more robust off the data manifold than on it, its input gradients are forced to lie approximately in the tangent bundle of the data manifold (Srinivas et al., 2024). This mechanism explains why robust models tend to have gradients that follow intrinsic manifold directions rather than adversarial spikes orthogonal to the data, and is consistent with the broader PAG literature where robust models exhibit gradients that align with human perceptual judgments (Ganz et al., 2023a; Kaur et al., 2019a; Ganz et al., 2023b).

**Composite pullback metric (layer aggregation).** Our composite perceptual metric is constructed by pulling back a Euclidean metric from a robust feature space through the generator. In differential-geometric terms this is the standard pullback construction: a Riemannian metric on the feature space induces a Riemannian metric on latent space via the generator map. Infinitesimal moves in the latent space $z$ are measured according to how they change robust features: directions along which robust representations vary smoothly and semantically incur low cost, while directions that robust training suppresses—such as high-frequency or adversarial perturbations—are assigned high cost. Geodesics under $g_{\mathcal{Z}}$ are therefore strongly biased to follow manifold-aligned, perceptually meaningful directions that the robust model uses internally, rather than the brittle directions preferred under Euclidean pixel distances or non-robust features. This picture mirrors the literature on robust perceptual similarity, which shows that distances in robust feature spaces correlate more strongly with human similarity judgments and are less vulnerable to adversarial manipulation than their standard counterparts (Kettunen et al., 2019; Ghazanfari et al., 2024a). Kettunen et al. demonstrate that LPIPS-type metrics are themselves vulnerable and can be rectified into a robust variant by using robust representations (Kettunen et al., 2019), while Ghazanfari et al. construct a 1-Lipschitz perceptual similarity metric with provable robustness guarantees and show that robustness in feature space improves both adversarial stability and alignment with human perceptual similarity (Ghazanfari et al., 2024a). Together with the manifold-alignment results above, these works support the view that infinitesimal moves in robust feature space correspond to smooth, semantically coherent deformations along the data manifold.

In practice, we instantiate the feature map as a composite feature map obtained by aggregating intermediate representations at multiple depths of an $L_2$-robust ResNet-50 with $\varepsilon = 3.0$ and, in the appendix, an $L_2$-robust XCiT-S12 backbone with the same threat radius. For the robust ResNet-50, we concatenate activations from the stem (layer 0) and all four residual stages (layers 1–4), and normalize each stage with simple scalar weights so that no single block dominates the metric. For the robust XCiT-S12, we follow the same principle and aggregate embeddings from early-to-mid blocks and mid-to-deep blocks, as well as a multi-block configuration that spans early and deep layers; features from different depths are rescaled and concatenated into a single representation. This composite robust pullback metric is the default geometry used by PCG. For comparison, we also define a "standard" pullback metric based on the same layer-aggregation scheme but using a standard (non-robust) ResNet-50 backbone. Across all backbones, the role of robustness is to supply feature spaces and gradients that are better aligned with the data manifold and human perception (Zaher et al., 2024; Srinivas et al., 2024; Ganz et al., 2023a; Kaur et al., 2019a; Ganz et al., 2023b; Kettunen et al., 2019; Ghazanfari et al., 2024a; Zhang & Zhu, 2019; Tsipras et al., 2019; Ilyas et al., 2019; Stutz et al., 2019), while the pullback construction translates this structure into a latent-space geometry that PCG can exploit.

---

**Algorithm 1** Perceptual Counterfactual Geodesics (PCG)

**Require:** Input image $x^\star$, target class $y'$, encoder $e$, generator $g$, classifier $f$
**Require:** Robust feature maps $\{h_k\}_{k=1}^K$, path length $T$, Phase-1 steps $S_1$, Phase-2 steps $S_2$
**Require:** Learning rate $\eta$, loss weight schedule $\{\lambda_s\}_{s=1}^{S_2}$, re-anchoring period $P$

1: **function** ROBUSTENERGY($\mathbf{z} = [z_0, \ldots, z_T]$)
2:     $\delta t \leftarrow 1/T, \quad E \leftarrow 0$
3:     **for** $i = 0$ to $T - 1$ **do**
4:         **for** $k = 1$ to $K$ **do**
5:             $u_{ik} \leftarrow h_k(g(z_{i+1})) - h_k(g(z_i))$
6:             $E \leftarrow E + \frac{1}{2} \frac{1}{\delta t} \|u_{ik}\|_2^2$
7:         **end for**
8:     **end for**
9:     **return** $E$
10: **end function**

11: **Initialization:**
12: $z_0 \leftarrow e(x^\star)$
13: Choose a target-class sample $x_{\text{tgt}}$ with $\arg\max f(x_{\text{tgt}}) = y'$
14: $z_T \leftarrow e(x_{\text{tgt}})$
15: Initialize $\{z_i\}_{i=1}^{T-1}$ by linear interpolation between $z_0$ and $z_T$

16: **Phase 1: Robust geodesic with fixed endpoints**
17: **for** $s = 1$ to $S_1$ **do**
18:     $E \leftarrow$ ROBUSTENERGY($[z_0, \ldots, z_T]$)
19:     Compute $\nabla_{z_1, \ldots, z_{T-1}} E$ by backprop
20:     **for** $i = 1$ to $T - 1$ **do**
21:         $z_i \leftarrow z_i - \eta \nabla_{z_i} E$
22:     **end for**
23: **end for**

24: **Phase 2: Endpoint-aware refinement under classification constraint**
25: **for** $s = 1$ to $S_2$ **do**
26:     $E \leftarrow$ ROBUSTENERGY($[z_0, \ldots, z_T]$)
27:     $\mathcal{L}_{\text{cls}} \leftarrow \ell(f(g(z_T)), y')$
28:     $\mathcal{L} \leftarrow E + \lambda_s \mathcal{L}_{\text{cls}}$
29:     Compute $\nabla_{z_1, \ldots, z_T} \mathcal{L}$ by backprop
30:     **for** $i = 1$ to $T - 1$ **do**
31:         $z_i \leftarrow z_i - \eta \nabla_{z_i} \mathcal{L}$
32:     **end for**
33:     $z_T \leftarrow z_T - \eta \nabla_{z_T} \mathcal{L}$                    ▷ endpoint update
34:     **if** $s \bmod P = 0$ **then**                    ▷ re-anchoring
35:         Re-anchor $z_T$ to the closest point along the path classified as $y'$
36:         Densify path by inserting midpoints and resampling to $T+1$ points
37:     **end if**
38: **end for**

39: **Return** final path $[z_0, \ldots, z_T]$ and counterfactual $x_{\text{cf}} = g(z_T)$

---

**Smoothness.** To ensure the induced metric varies smoothly, we replace non-smooth ReLU variants in our models with Softplus *post hoc* (after training). In practice this does not materially change behavior, as activations typically operate in smooth regions; it only guarantees that the metric field is differentiable along the paths we optimize.

### A.5 More Details on Evaluation Metrics

We briefly collect the definitions, motivations, and implementation details for all evaluation metrics used in the main text.

**FID and R-FID (realism).** To assess distribution-level realism, we use the Fréchet Inception Distance (FID), where lower values indicate that the generated distribution is closer to the real one in the chosen feature space.

In our setting, the real set comprises images from the target class in the training data, and the generated set comprises the corresponding target-class counterfactuals. As discussed in the main text, standard FID can be insensitive to non-robust or adversarial directions exploited by generative models. We therefore also report a robust variant, R-FID [], obtained by replacing the model used for FID with its robust. The functional form of the metric is identical; only the feature space changes. Intuitively, improvements that rely on non-robust directions tend to be reflected in standard FID but are penalised by R-FID, which is more tightly aligned with perceptual and semantic structure.

**LPIPS and R-LPIPS (closeness).** To quantify closeness between an original image $x_{\mathrm{orig}}$ and its counterfactual $x_{\mathrm{cf}}$ we use the Learned Perceptual Image Patch Similarity (LPIPS). Lower values under this metric correspond to smaller perceptual changes.

Analogously to FID/R-FID, we also use R-LPIPS that replaces the backbone with its robust counterpart. This yields a perceptual distance that is more stable under adversarial perturbations and better aligned with robust feature distance. In our context, LPIPS and R-LPIPS play complementary roles: they measure instance-level closeness between $x_{\mathrm{orig}}$ and $x_{\mathrm{cf}}$, whereas FID/R-FID capture distribution-level realism.

**COUT (representation-level sparsity).** Following the spirit of COUT from Khorram & Fuxin (2022), we measure how focused the change in the *explained* classifier's internal representation is relative to the change in its belief about the target class. Let $f : \mathcal{X} \to \mathbb{R}^C$ be the classifier we aim to explain, with logits $f(x)$ and target class $y_{\mathrm{tgt}}$, and let $h : \mathcal{X} \to \mathbb{R}^d$ denote a fixed internal representation of $f$ (penultimate layer). For a counterfactual pair $(x_{\mathrm{orig}}, x_{\mathrm{cf}})$, we define

$$\Delta f_{\mathrm{tgt}} = f_{y_{\mathrm{tgt}}}(x_{\mathrm{cf}}) - f_{y_{\mathrm{tgt}}}(x_{\mathrm{orig}}), \qquad \Delta h = h(x_{\mathrm{cf}}) - h(x_{\mathrm{orig}}). \tag{3}$$

The COUT score is then

$$\mathrm{COUT}(x_{\mathrm{orig}}, x_{\mathrm{cf}}) = \frac{\Delta f_{\mathrm{tgt}}}{\|\Delta h\|_2 + \epsilon}, \tag{4}$$

where $\epsilon$ is a small constant for numerical stability. Higher COUT indicates that the method obtains a given increase in target-class confidence with a smaller, more concentrated change in *the explained model's own* internal representation, which is desirable for counterfactual explanations that aim for targeted, rather than dispersed, changes. We adopt this representation-level analogue instead of the original pixel-curve COUT for two reasons. First, our setting is explicitly adversarially vulnerable: pixel-space perturbation curves can look favourable even when the underlying trajectory exploits non-robust directions, whereas measuring sparsity in $h$ probes how efficiently the counterfactual steers the actual decision-making features of $f$. Second, our framework is centred on feature and latent geometries rather than input-space masks, and a COUT defined in $h$ integrates more naturally with this viewpoint and is computationally lighter than evaluating per-pixel perturbation paths. Robustness and manifold faithfulness are handled separately by our semantic-margin and manifold-alignment diagnostics, which are computed in an independent robust feature space; COUT is thus read as an explanandum-relative efficiency measure that complements, rather than replaces, these robustness-aware metrics.

**Semantic margin (SM: semantic locality in robust feature space).** To probe whether counterfactuals move into regions of feature space genuinely associated with the target class, we use a semantic margin defined in a separate robust feature space. Let $\varphi : \mathcal{X} \to \mathbb{R}^D$ be a robust backbone (Inception-V3) that is *not* used to construct the PCG geometry. Let $\{(x_i, y_i)\}_{i=1}^N$ denote the labelled training set and $\varphi_i = \varphi(x_i)$.

For a counterfactual $x_{\text{cf}}$ with target class $y_{\text{tgt}}$, we define the sets of $k$ nearest neighbours in $\varphi$-space:

$$\mathcal{N}_{\text{tgt}}(x_{\text{cf}}) = \underset{\substack{\mathcal{S} \subset \{i:y_i = y_{\text{tgt}}\} \\ |\mathcal{S}| = k}}{\arg\min} \sum_{i \in \mathcal{S}} \left\| \varphi(x_{\text{cf}}) - \varphi_i \right\|_2, \tag{5}$$

$$\mathcal{N}_{\text{other}}(x_{\text{cf}}) = \underset{\substack{\mathcal{S} \subset \{i:y_i \neq y_{\text{tgt}}\} \\ |\mathcal{S}| = k}}{\arg\min} \sum_{i \in \mathcal{S}} \left\| \varphi(x_{\text{cf}}) - \varphi_i \right\|_2. \tag{6}$$

We then define class-conditional average distances

$$d_{\text{tgt}}(x_{\text{cf}}) = \frac{1}{k} \sum_{i \in \mathcal{N}_{\text{tgt}}(x_{\text{cf}})} \left\| \varphi(x_{\text{cf}}) - \varphi_i \right\|_2, \qquad d_{\text{other}}(x_{\text{cf}}) = \frac{1}{k} \sum_{i \in \mathcal{N}_{\text{other}}(x_{\text{cf}})} \left\| \varphi(x_{\text{cf}}) - \varphi_i \right\|_2, \tag{7}$$

and the semantic margin

$$m(x_{\text{cf}}) = d_{\text{other}}(x_{\text{cf}}) - d_{\text{tgt}}(x_{\text{cf}}). \tag{8}$$

Intuitively, $m(x_{\text{cf}}) > 0$ means that $x_{\text{cf}}$ is, on average, closer (in robust feature space) to target-class training examples than to non-target examples; $m(x_{\text{cf}}) \leq 0$ suggests that the counterfactual resides in a mixed or non-target neighbourhood and is therefore suspect from a manifold perspective (potentially off- or on-manifold AE). We report the mean semantic margin across all counterfactuals for each method in the main text. In our experiments, we set $k = 16$.

**Manifold Alignment Score (MAS: tangent alignment under different geometries).**    To quantify whether counterfactual updates follow the tangent structure induced by different geometries, we introduce a Manifold Alignment Score (MAS). For each geometry $g$ we consider a representation map

$$f_g : \mathcal{Z} \to \mathbb{R}^{p_g}, \tag{9}$$

obtained by composing the generator $G : \mathcal{Z} \to \mathcal{X}$ with an appropriate embedding:

$$f_{\text{pix}}(z) = G(z) \qquad \text{(pixel geometry)}, \tag{10}$$

$$f_{\text{std}}(z) = \psi\big(G(z)\big) \qquad \text{(standard feature geometry)}, \tag{11}$$

$$f_{\text{rob}}(z) = \Phi\big(G(z)\big) \qquad \text{(robust feature geometry)}. \tag{12}$$

Let $z_{\text{orig}}$ and $z_{\text{cf}}$ denote the latent codes of the original and counterfactual images, so that $x_{\text{orig}} = G(z_{\text{orig}})$ and $x_{\text{cf}} = G(z_{\text{cf}})$. For geometry $g$, we define the ambient displacement

$$v_g = \frac{f_g(z_{\text{cf}}) - f_g(z_{\text{orig}})}{\|f_g(z_{\text{cf}}) - f_g(z_{\text{orig}})\|_2^2} \in \mathbb{R}^{p_g}. \tag{13}$$

Using the Jacobian $J_g(z_{\text{orig}}) = \nabla_z f_g(z)\big|_{z=z_{\text{orig}}}$, we construct an orthogonal projector $P_g(z_{\text{orig}})$ onto the tangent space of the corresponding manifold at $f_g(z_{\text{orig}})$, for example via

$$P_g(z_{\text{orig}}) = J_g(z_{\text{orig}}) \big(J_g(z_{\text{orig}})^\top J_g(z_{\text{orig}})\big)^{-1} J_g(z_{\text{orig}})^\top. \tag{14}$$

The MAS for a pair $(x_{\text{orig}}, x_{\text{cf}})$ under geometry $g$ is then

$$s_g(x_{\text{orig}}, x_{\text{cf}}) = \frac{\left\| P_g(z_{\text{orig}}) v_g \right\|_2^2}{\|v_g\|_2^2} \in [0, 1]. \tag{15}$$

This score measures the fraction of the squared norm of the counterfactual displacement that lies in the tangent space induced by geometry $g$. High scores indicate that the update is predominantly tangent (manifold-aligned), whereas low scores indicate a large normal component (off-manifold or geometry-misaligned). We report averages of $s_g$ over all counterfactuals for each method and geometry.

**Path-based LPIPS and R-LPIPS (geodesic smoothness).**    To assess the smoothness of counterfactual trajectories, we use path-based perceptual metrics centred around the PCG geodesic and compare them to standard latent-space interpolations. Given a discrete path $\{x_t\}_{t=0}^{T}$ between $x_{\text{orig}}$ and $x_{\text{cf}}$ (PCG geodesic, linear latent path, or spherical latent path), we define the average LPIPS-step as

$$\overline{\Delta\text{LPIPS}} = \frac{1}{T} \sum_{t=0}^{T-1} \text{LPIPS}(x_t, x_{t+1}), \tag{16}$$

and analogously $\overline{\Delta\text{R-LPIPS}}$ by replacing LPIPS with R-LPIPS. Smaller values indicate more gradual perceptual change along the path.

# B MORE RESULTS & ANALYSIS

## B.1 INTERPOLATION RESULTS

Figures 5 and 6 compare straight-line interpolations under four geometries based on STYLEGAN3. From top to bottom in each panel: (i) $Z$-linear interpolation (flat latent space), (ii) pixel-space MSE pullback ($\mathcal{X}_{\mathrm{MSE}}$), (iii) standard feature pullback ($\mathcal{F}_{\mathrm{MSE}}$), and (iv) our robust perceptual pullback ($\mathcal{R}_{\mathrm{MSE}}$). The robust metric produces smooth, on-manifold transitions with consistent semantics (identity/pose for faces; class coherence for animals), while $Z$-lerp and pixel MSE exhibit mid-trajectory artifacts and blends. The standard feature pullback improves semantics but still suffer from similar failure modes. These visuals mirror the trends discussed in the main text and motivate using a robust geometry for PCG.

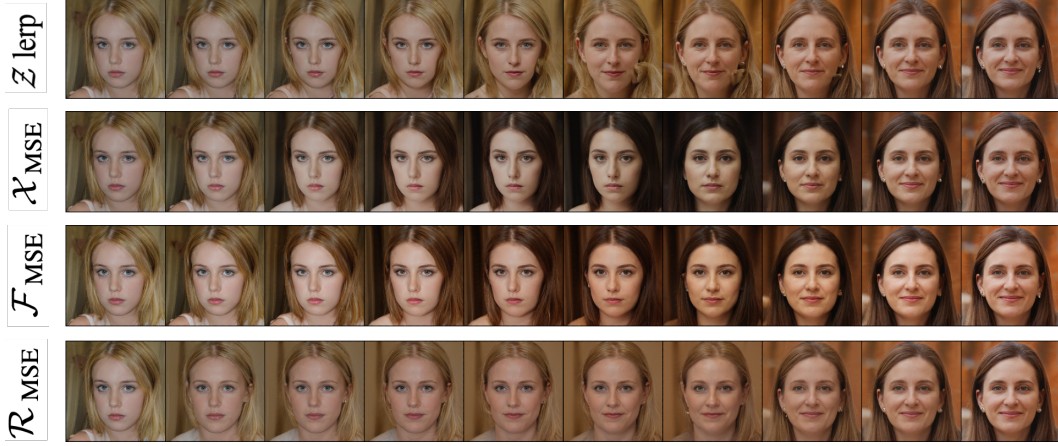

Figure 5: Interpolations on FFHQ under four geometries. Rows (top to bottom): $Z$-lerp, $\mathcal{X}_{\mathrm{MSE}}$ pullback, $\mathcal{F}_{\mathrm{MSE}}$ pullback, and robust $\mathcal{R}_{\mathrm{MSE}}$ pullback. The robust row shows a smooth, semantically consistent evolution (e.g., gradual attribute change without identity drift), whereas the other geometries introduce off-manifold blends and texture/illumination artifacts mid-path.

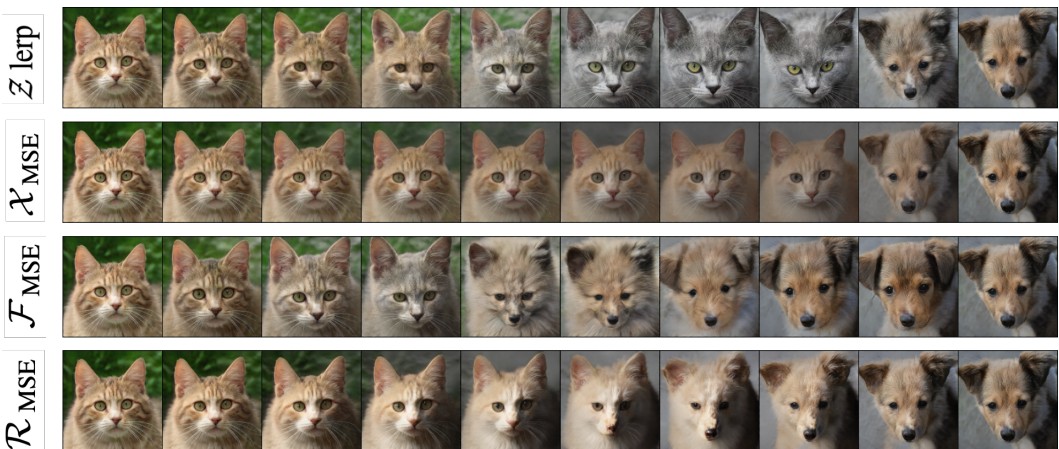

Figure 6: Interpolations on AFHQ under four geometries. Same ordering as Fig. 5. The robust $\mathcal{R}_{\mathrm{MSE}}$ path preserves class coherence and yields clean transitions, while $Z$-lerp and $\mathcal{X}_{\mathrm{MSE}}$ produce ambiguous hybrids and brittle textures; $\mathcal{F}_{\mathrm{MSE}}$ reduces but does not eliminate these effects.

## B.2 Perceptual Counterfactual Geodesics across AFHQ and FFHQ.

Figures 7 (AFHQ, two examples) and 8 (FFHQ, two examples) visualize the two-phase PCG procedure with STYLEGAN3. In each panel, the top row is the initial linear path in $Z$ (straight interpolation between the encoded input and a target exemplar), which often drifts off-manifold or blends semantics mid-trajectory. The middle row is the Phase 1 robust geodesic with fixed endpoints; transitions become smooth and class-consistent. The bottom row is the Phase 2 counterfactual geodesic, where the endpoint is jointly refined with the classification loss; the endpoint moves closer to the input while achieving the target class/attribute, and the entire path remains on-manifold. Qualitatively, AFHQ preserves species structure and textures, while FFHQ preserves identity and pose as attributes change, supporting the claims about semantic fidelity and geometry-aware paths.

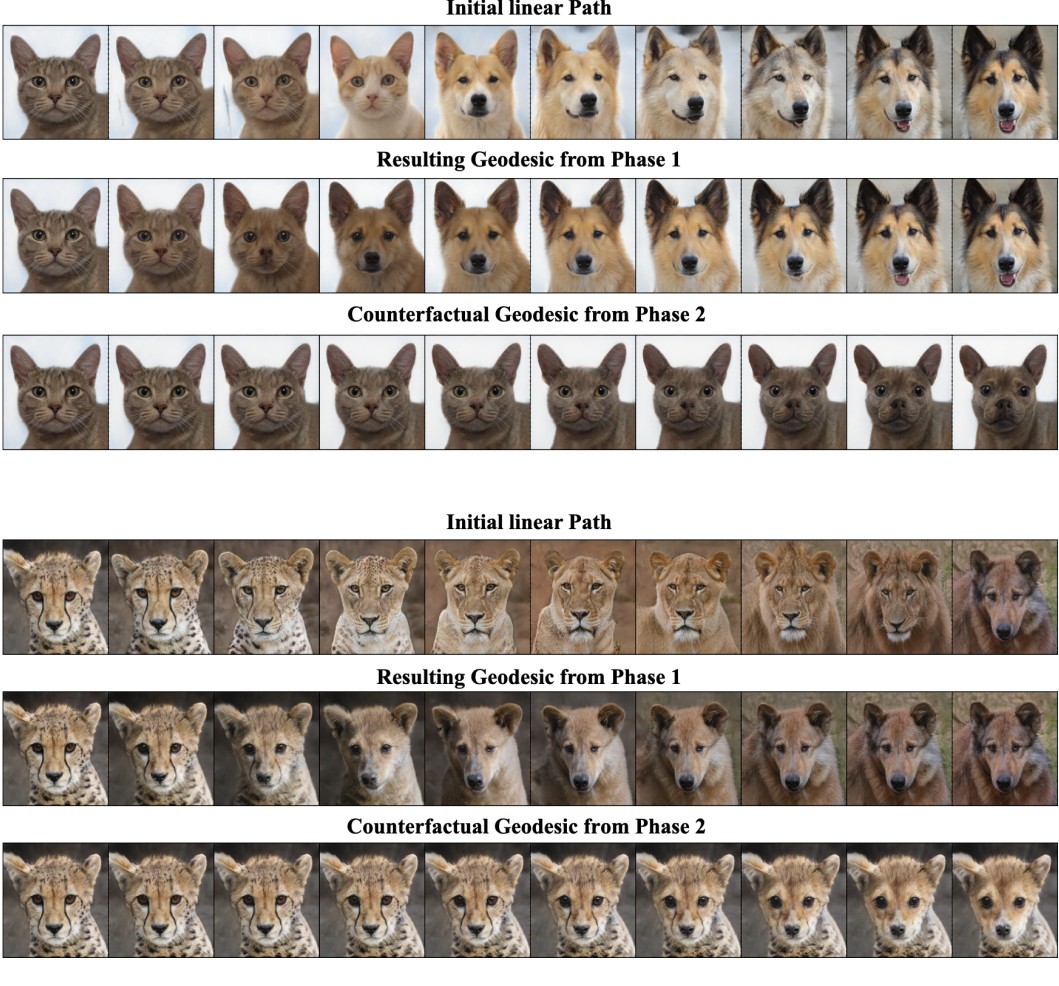

Figure 7: PCG on AFHQ (STYLEGAN3), two examples (Cat → Dog & Wild → Dog). Rows (top to bottom): initial linear path in $Z$ between the encoded input and a target exemplar; Phase 1 robust geodesic (energy-only) with fixed endpoints; Phase 2 counterfactual geodesic after endpoint refinement with classification loss. The geodesic rows remove mid-path blends and keep species-level semantics while reaching the target class.

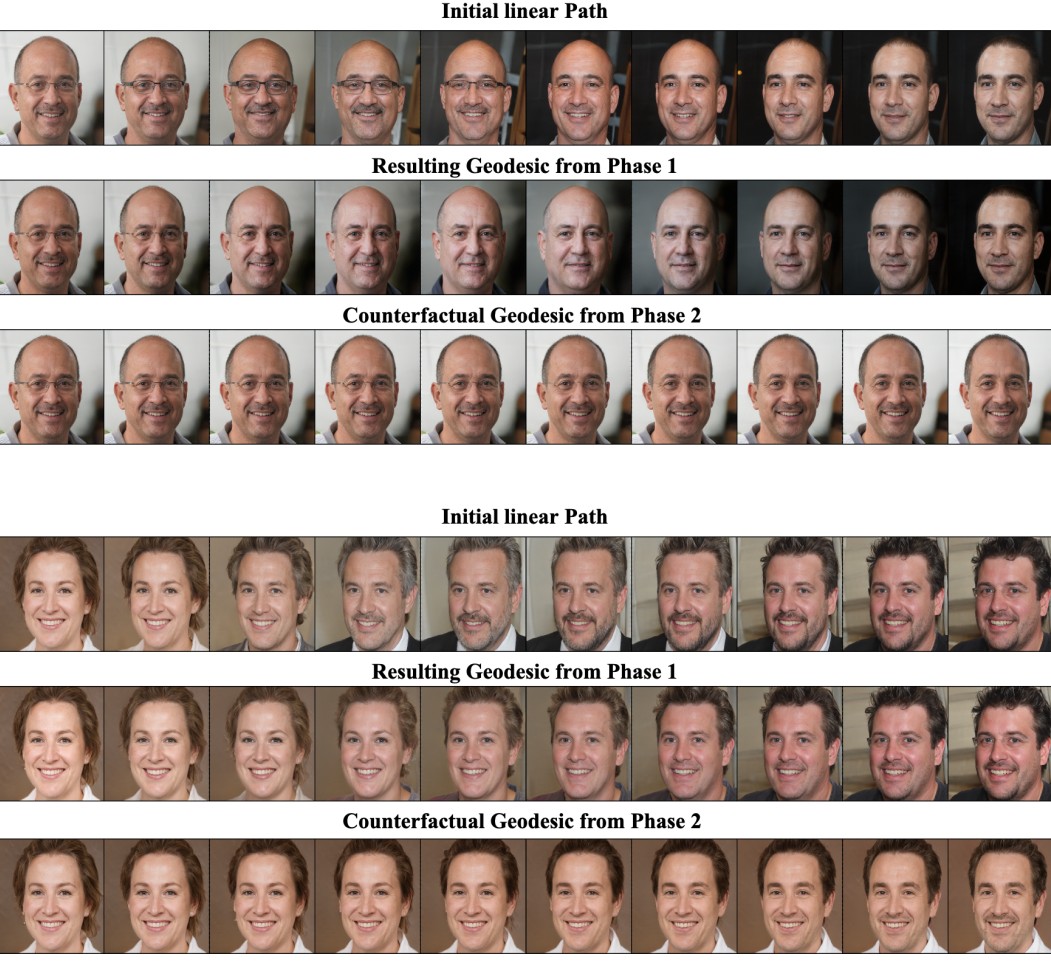

Figure 8: PCG on FFHQ (StyleGAN3), two examples (Glasses → No-glasses & Female → Male). Same layout as Fig. 7. Phase 1 produces smooth, on-manifold transitions; Phase 2 moves the endpoint toward the input while satisfying the target classifier. Identity and pose are largely preserved as the target attribute changes, and intermediate frames remain perceptually coherent.

### B.3 SENSITIVITY TO DIFFERENT TARGET CLASS SAMPLES

Figures 9 and 10 test how PCG depends on which target-class exemplar is used to initialize the path. For each input we run PCG twice, once per exemplar. We observe that the Phase 1 geodesic reflects the chosen exemplar (different coarse routes in latent space), but after Phase 2 (endpoint refinement with classification loss) the counterfactual geodesics converge to a tight neighborhood around the input while achieving the target label/attribute. This yields diverse yet faithful counterfactuals and supports the main-text claim about robustness to target initialization.

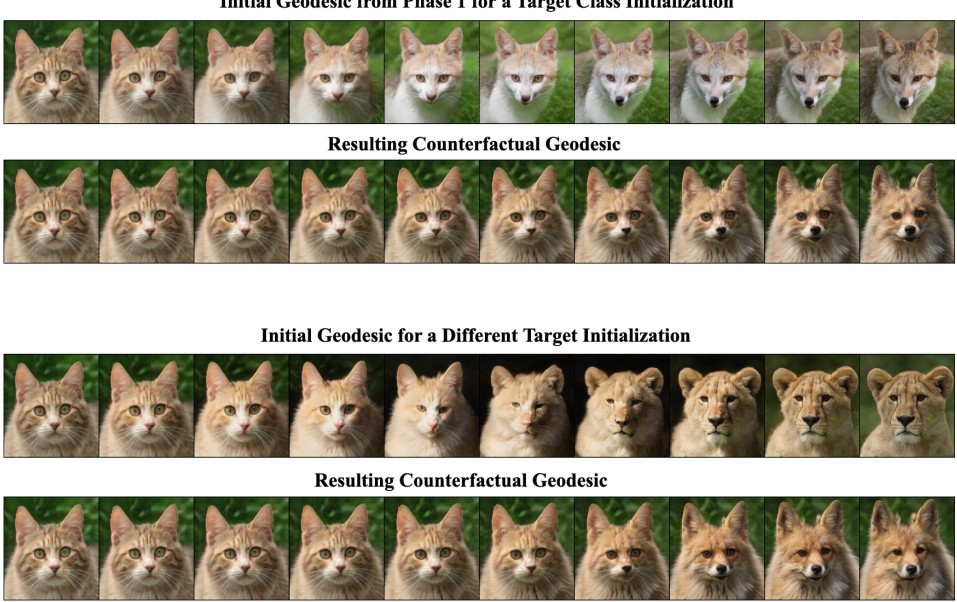

Figure 9: Sensitivity to target exemplar on AFHQ (StyleGAN3) (Cat → Wild). Rows: (1) Phase 1 geodesic initialized with target exemplar A, (2) resulting Phase 2 counterfactual geodesic, (3) Phase 1 geodesic with a different exemplar B, (4) resulting Phase 2 counterfactual geodesic. Although the Phase 1 routes differ, the Phase 2 counterfactuals converge near the input and satisfy the target class, indicating low sensitivity to the exemplar choice and producing diverse but faithful variations.

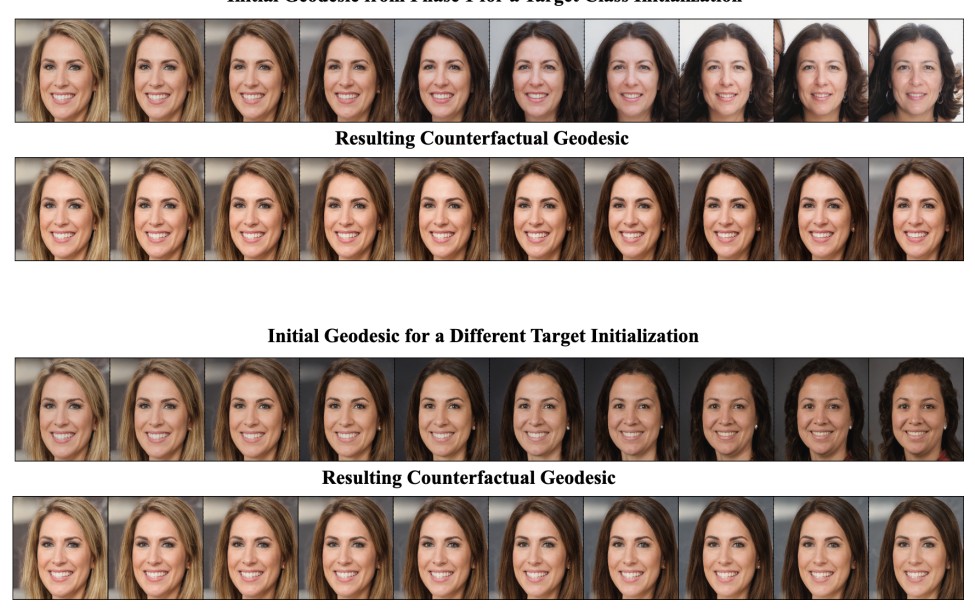

Figure 10: Sensitivity to target exemplar on FFHQ (StyleGAN3) (Blonde → Non-blonde). Same layout as Fig. 9. Two different target exemplars lead to distinct Phase 1 paths, yet the Phase 2 counterfactual geodesics converge to a small neighborhood around the input while achieving the target attribute; identity and pose remain largely preserved.

**Quantitative sensitivity (multiple initializations, $M{=}15$ per input).** To quantify low sensitivity to target initialization, we run PCG $M{=}15$ times per input with different target exemplars and measure

how close and consistent the resulting counterfactuals (CFs) are. We use LPIPS and its robust variant R-LPIPS, and report three intuitive, scale-aware metrics: (i) **CF dispersion ratio (CDR)** — how tightly CFs cluster compared to typical variation within the target class, (ii) **Initialization–output contraction ratio (IOCR)** — how strongly optimization contracts the diversity of target initializations into a tight CF cluster, and (iii) **CF diameter** — the worst-case dissimilarity among the CFs. All three metrics are computed both in standard LPIPS space and in the robust feature space underlying R-LPIPS.

**Definitions.** Let $C = \{x_{\text{cf}}^{(1)}, \ldots, x_{\text{cf}}^{(M)}\}$ be the CF endpoints for one input $x^\star$, with $M = 15$. Let $T = \{x_{\text{tgt}}^{(1)}, \ldots, x_{\text{tgt}}^{(M)}\}$ denote the corresponding target-class initializations used to initialise the path endpoints. Let $d(\cdot, \cdot)$ denote a perceptual distance, instantiated either as LPIPS or as R-LPIPS. Let $\overline{d}_{\text{tgt}}$ be the average $d$-distance between random pairs sampled from the *target* class (estimated once per dataset/attribute using 30 random pairs).

*(1) CDR (CF dispersion ratio):*

$$\overline{d}_{\text{CF}} = \frac{2}{M(M-1)} \sum_{m<n} d\big(x_{\text{cf}}^{(m)}, x_{\text{cf}}^{(n)}\big), \qquad \text{CDR} = \frac{\overline{d}_{\text{CF}}}{\overline{d}_{\text{tgt}}}.$$

$\text{CDR} \ll 1$ indicates CFs form a cluster much tighter than typical target-class variability.

*(2) IOCR (Initialization–output contraction ratio):* First, measure the average dispersion of the *initial* target exemplars:

$$\overline{d}_{\text{init}} = \frac{2}{M(M-1)} \sum_{m<n} d\big(x_{\text{tgt}}^{(m)}, x_{\text{tgt}}^{(n)}\big).$$

We then define

$$\text{IOCR} = \frac{\overline{d}_{\text{CF}}}{\overline{d}_{\text{init}}}.$$

Here $\text{IOCR} \ll 1$ means that optimization contracts a diverse set of target initializations into a much tighter CF cluster (strong insensitivity to the choice of target exemplar), whereas $\text{IOCR} \approx 1$ indicates that CFs are about as diverse as the initial targets.

*(3) CF diameter:*

$$\text{Diam}_{\text{CF}} = \max_{m<n} d\big(x_{\text{cf}}^{(m)}, x_{\text{cf}}^{(n)}\big),$$

so a small value guarantees even the most dissimilar CFs among the $M$ runs remain close. We report all three quantities under both $d = \text{LPIPS}$ and $d = \text{R-LPIPS}$ to jointly capture sensitivity in standard and robust perceptual feature spaces.

**Sensitivity summary (AFHQ and FFHQ) based on STYLEGAN2.** Using the LPIPS-based metrics defined above, Table 5 reports the *CF dispersion ratio* (CDR), the *Initialization–output contraction ratio* (IOCR), and the *CF diameter* for $M=15$ target initializations per input (mean $\pm$ std). For each dataset/task, CDR is the intra-CF mean perceptual distance normalised by a target-class baseline $\overline{d}_{\text{tgt}}$ computed from 30 random target-class pairs; by construction, $\text{CDR} = 1$ corresponds to the variability of two random target-class samples, while $\text{CDR} \ll 1$ indicates that CFs form a much tighter cluster than generic target-class variation. IOCR compares the intra-CF dispersion to the dispersion of the initial target exemplars for that input; here $\text{IOCR} = 1$ means the CFs are about as diverse as the initial targets, whereas $\text{IOCR} \ll 1$ indicates that optimization contracts a diverse set of initial targets into a tighter CF neighborhood. CF diameter is the maximum pairwise distance among the 15 CFs and captures the worst-case gap within the cluster. All three metrics are evaluated both in standard LPIPS space and in the robust R-LPIPS feature space. In all cases, *lower is better*: tighter clustering (CDR), stronger contraction of initial diversity (IOCR), and smaller worst-case separation (diameter).

For all AFHQ and FFHQ tasks, we observe $\text{CDR} \ll 1$, showing that CFs produced from different target exemplars lie in a cluster that is several times tighter than typical target-class variability, consistent with our claim that PCG converges on a stable, input-specific counterfactual neighborhood rather than scattering across the class. IOCR values well below 1 further indicate that optimization strongly contracts the diversity of the initial target exemplars into this neighborhood, i.e. the final counterfactual is largely insensitive to which target exemplar was used. The small CF diameters

confirm this even in the worst case: the most dissimilar CFs remain close in both standard and robust perceptual feature spaces. The same trends hold under LPIPS and its robust counterpart R-LPIPS (denoted LPIPS and R-LPIPS in the table), indicating that this insensitivity is preserved when measured in a robustness-aware feature space.

Table 5: Sensitivity to target initialization (15 runs per input).

| Task | CDR | | IOCR | | CF Diam. | |
|---|---|---|---|---|---|---|
| | LPIPS | R-LPIPS | LPIPS | R-LPIPS | LPIPS | R-LPIPS |
| AFHQ: cat → dog | $0.21 \pm 0.08$ | $0.16 \pm 0.05$ | $0.23 \pm 0.03$ | $0.17 \pm 0.08$ | $0.17 \pm 0.03$ | $0.15 \pm 0.04$ |
| FFHQ: not-smile → smile | $0.22 \pm 0.05$ | $0.18 \pm 0.07$ | $0.28 \pm 0.02$ | $0.21 \pm 0.03$ | $0.13 \pm 0.05$ | $0.12 \pm 0.06$ |
| FFHQ: bald → hairy | $0.31 \pm 0.07$ | $0.24 \pm 0.06$ | $0.25 \pm 0.05$ | $0.21 \pm 0.09$ | $0.21 \pm 0.02$ | $0.17 \pm 0.05$ |

## B.4 QUANTITATIVE RESULTS BASED ON STYLEGAN3

As in the main text, PCG consistently achieves the lowest values under the geometry-aware metrics $\mathcal{L}_{\mathcal{F}}$ and $\mathcal{L}_{\mathcal{R}}$ and remains competitive under pixel metrics. These appendix results, obtained on STYLEGAN3, show that the robust geodesic formulation retains its advantage without re-tuning and confirm the stability of PCG's behaviour across model choices.

Table 6: Quantitative comparison across datasets.

| Method | AFHQ | | | | FFHQ | | | |
|---|---|---|---|---|---|---|---|---|
| | $\mathcal{L}_1$ | $\mathcal{L}_2$ | $\mathcal{L}_{\mathcal{F}}$ | $\mathcal{L}_{\mathcal{R}}$ | $\mathcal{L}_1$ | $\mathcal{L}_2$ | $\mathcal{L}_{\mathcal{F}}$ | $\mathcal{L}_{\mathcal{R}}$ |
| REVISE | 1.18±0.12 | **0.72**±0.17 | 1.05±0.10 | 2.68±0.04 | 0.81±0.07 | **0.33**±0.12 | 0.81±0.09 | 2.75±0.06 |
| VSGD | 1.30±0.11 | 1.48±0.15 | 1.57±0.09 | 2.88±0.08 | 0.78±0.11 | 0.95±0.10 | 1.49±0.12 | 2.83±0.08 |
| RSGD | 0.84±0.08 | 1.30±0.09 | 0.68±0.07 | 1.83±0.05 | 0.60±0.05 | 0.83±0.07 | 0.60±0.04 | 2.39±0.05 |
| RSGD-C | 0.92±0.10 | 1.43±0.16 | 0.63±0.08 | 1.73±0.06 | 0.67±0.06 | 0.91±0.09 | 0.47±0.04 | 2.08±0.05 |
| PCG (ours) | **0.78**±0.07 | 1.13±0.10 | **0.51**±0.06 | **0.30**±0.02 | **0.41**±0.03 | 0.71±0.09 | **0.38**±0.05 | **0.21**±0.05 |

## B.5 RUNTIME COMPLEXITY & SCALABILITY ON AFHQ

On AFHQ, measured on a single NVIDIA H100 GPU, Table 7 reports per-sample wall-clock runtimes and speedups across methods based on STYLEGAN2. VSGD is the fastest (1.6 min). PCG runs in 3.4 min per sample despite being path-based (here $T=10$): with a GPU, all path nodes and robust-feature evaluations are batched in a single forward/backward, so the extra cost is modest. RSGD is slowest (5.7 min) because each step requires solving $G_Z(z)\, r = \nabla_z \mathcal{L}$ with Conjugate Gradients; the inner CG iterations and repeated Jacobian–vector products through $g$ (and, for RSGD-C, the feature backbone) dominate wall-clock. Absolute times depend on precision and batch sizing, but the relative ordering was consistent across runs.

Table 7: AFHQ per-sample wall-clock runtime (minutes). RSGD serves as a representative for RSGD/RSGD-C; VSGD represents standard Euclidean-gradient methods.

| Method | Time (min) | Speedup vs RSGD | Notes |
|---|---|---|---|
| VSGD (rep. Euclidean) | 1.6 | 3.56x | Classification loss only; lowest cost. |
| PCG (ours) | 3.4 | 1.68x | Path-based with $T=10$ nodes; nodes batched on GPU. |
| RSGD (rep. RSGD/–C) | 5.7 | 1.00x | Natural-gradient via CG; Jacobian–vector products dominate. |

**Trade-off between Path Length and Smoothness**. PCG parameterizes a counterfactual trajectory as a discretized path with $T$ points between $x_{\text{orig}}$ and $x_{\text{cf}}$. Increasing $T$ refines the discretization of the underlying robust geodesic: more points allow the optimizer to distribute semantic change across more intermediate states, but also increase compute and memory usage because each point carries its own latent code, generator activations, and robust features.

**Scaling of time and memory with $T$.** Table 8 reports runtime and peak CUDA memory for $512 \times 512$ images on a single GPU as we vary $T \in \{10, 15, 20, 25\}$. Empirically, wall-clock time grows approximately linearly with $T$: going from $T{=}10$ to $T{=}20$ roughly doubles path length and increases runtime from 3.4 to 7.2 minutes ($\sim 2.1\times$), while $T{=}25$ yields 10.9 minutes ($\sim 3.2\times$ the cost of $T{=}10$ for a $2.5\times$ longer path). This reflects the fact that each additional node contributes its own set of forward and backward passes through the generator and robust backbone. Peak CUDA memory also increases with $T$, but more gently: from 23.4GB at $T{=}10$ to 27.3GB at $T{=}25$ (an increase of $\sim 17\%$). Here, peak memory reflects the joint footprint of all components needed for our setup (pretrained generator and encoder, the classifier to be explained, the robust backbone used to induce geometry, and the activations and latents required to run PCG), not just the path itself.

**Scaling of time and memory with image resolution.** At fixed path length ($T{=}10$), PCG also scales smoothly across image resolutions (Table 9). Runtime increases roughly in line with the number of pixels: moving from $256^2$ to $512^2$ images doubles the resolution in each dimension and approximately doubles wall-clock time (from 1.7 to 3.4 minutes), while going from $512^2$ to $1024^2$ yields a further $\sim 2.3\times$ increase (from 3.4 to 7.8 minutes). Peak CUDA memory grows more moderately—from 19.8GB at $256^2$ to 23.4GB at $512^2$ and 27.7GB at $1024^2$—because a substantial fraction of the footprint comes from resolution-independent components (latents, robust features, and model parameters), with higher resolutions mainly contributing larger generator and classifier activations in the ambient image space. In practice, this means PCG remains feasible up to $1024^2$ on a single 24–40GB GPU, with runtime being the primary limiting factor at very high resolutions rather than memory exhaustion.

**Effect of $T$ on perceptual smoothness.** To quantify the smoothness side of this trade-off, we measure the path-based average LPIPS-step and its robust counterpart, $\overline{\Delta\text{LPIPS}}$ and $\overline{\Delta\text{R-LPIPS}}$ (see Appendix A.5 for definitions), for different path lengths $T \in \{3, 5, 8, 10, 15, 20\}$ while keeping all other hyperparameters fixed. The resulting values are visualized in Figure 11. We observe a clear pattern: very short paths ($T{=}3$ and $T{=}5$) exhibit noticeably larger $\overline{\Delta\text{LPIPS}}$ and $\overline{\Delta\text{R-LPIPS}}$, indicating that the class change is implemented via a small number of relatively large perceptual jumps. Increasing the path length to $T{=}8$ and $T{=}10$ yields a substantial reduction in both metrics, as the optimizer can resolve the robust geodesic with a finer discretization and distribute semantic change more evenly along the trajectory. Beyond $T{=}10$, the gains become marginal: $T{=}15$ provides a small additional improvement in $\overline{\Delta\text{LPIPS}}$ and $\overline{\Delta\text{R-LPIPS}}$, and $T{=}20$ essentially plateaus, with only minor decreases in step size despite the increased cost.

**Practical operating range and comparison to RSGD.** Taken together, the runtime and path-smoothness measurements provide a concrete view on the trade-off. Short paths are computationally cheap but produce perceptually coarser trajectories; very long paths yield only slightly smoother geodesics at a near-linear increase in time and a non-trivial increase in peak memory. In our experiments, $T{=}8$–$10$ emerges as a practical operating range: it captures the majority of the smoothness gains observed when increasing $T$, while keeping runtime and memory within a comfortable budget on a single high-memory GPU. When resources permit and extremely fine-grained trajectories are desired, $T{=}15$–$20$ can be used, but beyond this range we expect the cost to dominate the marginal improvements in $\overline{\Delta\text{LPIPS}}$ and $\overline{\Delta\text{R-LPIPS}}$. For reference, at $T{=}10$ and $512^2$ resolution, PCG runs in 3.4 minutes with a peak of 23.4GB, compared to RSGD/RSGD-C, which require 5.7 minutes and $\sim 29$GB under the same conditions due to repeated conjugate-gradient solves and Jacobian–vector products for metric inversion. VSGD, by contrast, is fastest (1.6 minutes, no path structure), but lacks any geometric regularization.

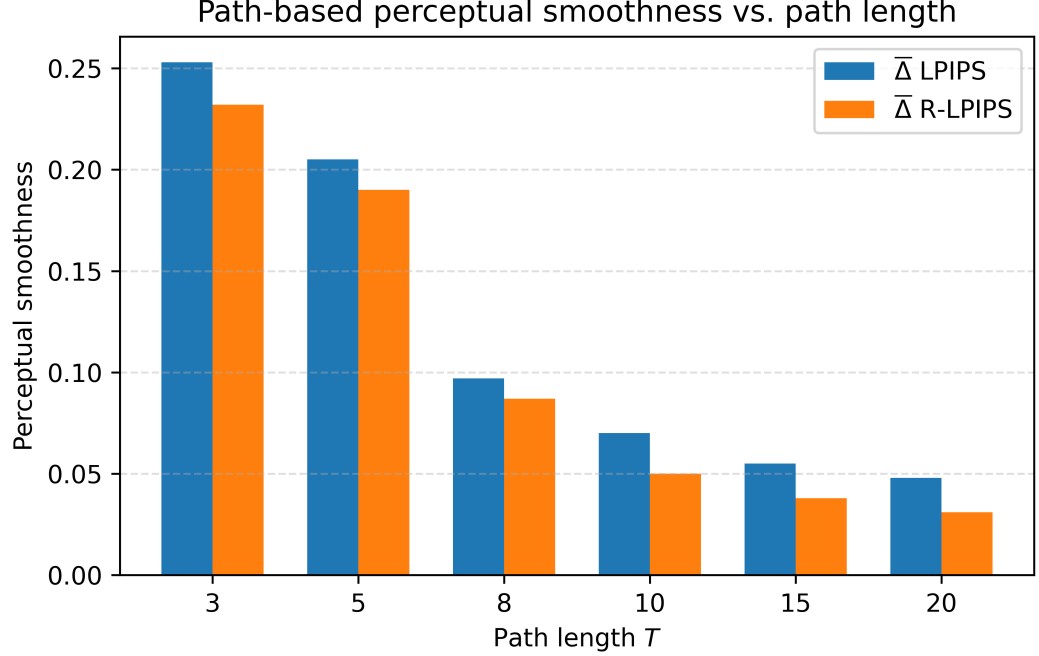

Figure 11: Path-based perceptual smoothness versus path length T, measured by average LPIPS and R-LPIPS step (lower is better).

Table 8: Resource usage across different path lengths (resolution $512^2$).

| Resources | Path Length $T$ | | | |
|---|---|---|---|---|
| | 10 | 15 | 20 | 25 |
| Time (min) | 3.4 | 5.3 | 7.2 | 10.9 |
| Peak CUDA Memory (GB) | 23.4 | 24.1 | 25.5 | 27.3 |

Table 9: Resource usage across different image resolutions ($T{=}10$ points).

| Resources | Image Resolution | | |
|---|---|---|---|
| | $256^2$ | $512^2$ | $1024^2$ |
| Time (min) | 1.7 | 3.4 | 7.8 |
| Peak CUDA Memory (GB) | 19.8 | 23.4 | 27.7 |

### B.6 ABLATIONS ON THE CLASSIFIER WEIGHT $\lambda$ FOR PCG

We study how PCG's behavior depends on the classifier-loss weight $\lambda$ under a fixed optimization budget of 300 steps. Recall that PCG initializes each path with an endpoint sampled from the target class, so the classifier already predicts the target at the start of optimization. To quantify whether optimization preserves this, we report a *target-class retention* metric:

$$\text{Target retention} = \frac{1}{N}\sum_{i=1}^{N}\mathbb{I}\big[f(x_{\text{end}}^{(i)}) = y_{\text{tgt}}^{(i)}\big],$$

where $x_{\text{end}}^{(i)}$ is the final endpoint after optimization and $y_{\text{tgt}}^{(i)}$ is the prescribed target class. High values mean that the endpoint remains in the target region; low values indicate that the path has lapsed back towards the source class.

We compute FID, R-FID, LPIPS, and R-LPIPS on a *shared* subset of examples: we first filter, for each configuration, to those counterfactuals whose endpoint is in the target class, and then take the intersection across all configurations. All reported distributional and perceptual metrics are evaluated on this common subset.

Table 10 reports results for a set of static $\lambda$ values (kept constant throughout optimization) and for dynamic schedules of the form $\lambda_t = \lambda_0 \cdot 5^{\lfloor t/50 \rfloor}$.

Table 10: Effect of the classifier-loss weight $\lambda$ on PCG under a 300-step budget. Static settings keep $\lambda$ fixed; dynamic schedules start from $\lambda$ and multiply by 5 every 50 steps. Lower is better for FID, R-FID, LPIPS, and R-LPIPS; higher is better for Target retention. (STYLEGAN2 on AFHQ)

| $\lambda$ **type** | **Setting** | **FID $\downarrow$** | **R-FID $\downarrow$** | **LPIPS $\downarrow$** | **R-LPIPS $\downarrow$** | **Target retention $\uparrow$** |
|---|---|---|---|---|---|---|
| Static | 0.0001 | 11.8 | 14.9 | 0.05 | 0.03 | 0.37 |
| | 0.005 | 10.3 | 11.2 | 0.07 | 0.04 | 0.54 |
| | 0.010 | 9.1 | 9.5 | 0.25 | 0.21 | 0.68 |
| | 0.100 | 8.5 | 9.3 | 0.31 | 0.27 | 0.78 |
| | 1.000 | 7.8 | 8.6 | 0.42 | 0.38 | 0.81 |
| | 4.000 | 7.4 | 8.2 | 0.51 | 0.41 | 0.87 |
| Dynamic | $\lambda_0 = 0.0001$ | 8.4 | 9.5 | 0.21 | 0.13 | 0.95 |
| | $\lambda_0 = 0.001$ | 7.9 | 9.8 | 0.47 | 0.32 | 0.96 |
| | $\lambda_0 = 0.01$ | 7.5 | 9.1 | 0.51 | 0.47 | 0.97 |

For very small static $\lambda$ (e.g. 0.0001), the path-energy term dominates the objective, so the optimiser prefers trajectories that remain extremely close to the original point. This yields low LPIPS / R-LPIPS at a higher FID, but the target-class retention is poor: the endpoint is often pulled back towards the source-class region within the 300-step budget, so the initial target-class endpoint is not maintained.

As $\lambda$ increases, target retention rises monotonically: moderate values (around 0.01–0.1 and up to 1.0) strike a better balance between path energy and classifier loss, achieving substantially higher retention while keeping FID and R-FID near their best values and only moderately increasing LPIPS. Pushing $\lambda$ to larger static values (e.g. $\lambda = 4.0$) further increases retention but at the cost of noticeably higher LPIPS / R-LPIPS, reflecting more aggressive, less conservative moves away from the original image.

Dynamic schedules achieve even higher target retention under the same budget. Starting from a small $\lambda$ and growing it multiplicatively allows early iterations to prioritize finding a low-energy path structure, while later iterations increasingly emphasize staying in the target-class region. This yields retention above 0.9 with FID / R-FID comparable to the best static settings. However, as the initial weight $\lambda$ increases (e.g. starting from 0.001 or 0.01), the classifier term dominates earlier in optimization: target retention approaches 0.96–0.97, but LPIPS and R-LPIPS increase, indicating relatively larger semantic displacement from the input. Overall, these trends highlight the intended trade-off: smaller $\lambda$ values favor geodesics that cling too tightly to the original image and often lose the target class at the endpoint, whereas larger and scheduled $\lambda$ values improve target-class retention at the expense of closeness, with moderate dynamic schedules offering the best compromise between geodesic regularity and counterfactual validity.

### B.7 SMOOTHNESS OF THE GENERATED COUNTERFACTUAL GEODESICS.

Table 11 compares different paths connecting the same endpoints, $x_{\text{orig}}$ and $x_{\text{cf}}$, in terms of path-based perceptual smoothness. For each trajectory, we compute the average LPIPS-step $\overline{\Delta\text{LPIPS}}$ and its robust counterpart $\overline{\Delta\text{R-LPIPS}}$ between consecutive points along the path. Both latent-space baselines—linear and spherical interpolation—exhibit relatively large $\overline{\Delta\text{LPIPS}}$ and $\overline{\Delta\text{R-LPIPS}}$, indicating that the perceptual change between successive frames is uneven and occasionally abrupt, particularly when measured in robust feature space. In contrast, the PCG geodesic achieves markedly smaller values of both $\overline{\Delta\text{LPIPS}}$ and $\overline{\Delta\text{R-LPIPS}}$. This is consistent with the construction of PCG as a discrete approximation to a geodesic in the induced robust perceptual geometry: by explicitly minimizing path energy in that metric, PCG produces trajectories that change content more gradually

and at approximately constant "perceptual speed" in robust feature space, whereas naive latent interpolations do not respect this geometry and therefore yield less regular, less semantically smooth transitions.

Table 11: Path-based perceptual smoothness for different interpolation schemes between $x_{\mathrm{orig}}$ and $x_{\mathrm{cf}}$. We report the average LPIPS and R-LPIPS step along each path, $\overline{\Delta\mathrm{LPIPS}}$ and $\overline{\Delta\mathrm{R\text{-}LPIPS}}$ (lower is better) (STYLEGAN2 on AFHQ).

| Path type | $\overline{\Delta\mathrm{LPIPS}} \downarrow$ | $\overline{\Delta\mathrm{R\text{-}LPIPS}} \downarrow$ |
|---|---|---|
| Linear latent interpolation | 0.51 | 0.83 |
| Spherical latent interpolation | 0.48 | 0.64 |
| PCG geodesic (ours) | **0.07** | **0.05** |

## B.8 EFFECTS OF THE CHOICE OF ROBUST BACKBONE AND AGGREGATED LAYERS.

Table 12 investigates how the induced geometry depends on the choice of robust backbone (CNN vs. Vision Transformers) and on which layers are aggregated. In all cases, the ResNet-50 and XCiT-S12 backbones are adversarially trained under an $\ell_2$ threat model with $\varepsilon = 3$ from (Engstrom et al., 2019; Debenedetti et al., 2023). For each backbone, we consider three aggregation configurations. *Early-to-mid* captures low-level and intermediate structure from early-to-mid layers/blocks (ResNet-50: stem and conv2_x; XCiT-S12: blocks 3 and 5). *Mid-to-deep* focuses on higher-level semantics from later layers/blocks (ResNet-50: conv3_x–conv5_x; XCiT-S12: blocks 7, 9, and 11). Finally, *multi-block* (our default configuration) aggregates features across early-to-deep layers.

Two patterns are clear. First, within each backbone, there is a consistent progression from early-to-mid, to mid-to-deep, to multi-block aggregation: R-FID decreases, while both MAS and mean SM increase. For the robust ResNet-50, moving from early-to-mid to mid-to-deep layers yields a gain in robust realism and semantic structure, and aggregating across all blocks further improves all three metrics. The XCiT-S12 backbone shows the same behaviour: early-only aggregation underperforms, mid-to-deep layers bring a clear improvement, and multi-block aggregation gives the strongest overall performance, albeit with slightly lower absolute values than ResNet-50. Second, across architectures, the relative trends and performance gaps remain stable: regardless of whether the robust features come from a CNN or a vision transformer, PCG benefits from including higher-level, semantically richer layers, and combining early and deep features yields the best trade-off between distributional realism (R-FID), manifold alignment (MAS), and semantic locality (mean SM). Note that, even with a few early layers across different backbones, PCG outperforms all baselines (refer to Table 2). This further shows that the proposed induced geometry is not tied to a specific robust backbone, while also highlighting that robust, high-level features are particularly important for capturing manifold structure.

Table 12: Effect of robust backbone and layer aggregation on PCG (STYLEGAN2 on AFHQ).

| Backbone | Layer set | R-FID $\downarrow$ | MAS $\uparrow$ | Mean SM $\uparrow$ |
|---|---|---|---|---|
| ResNet-50 | Early-to-mid | 12.2 | 0.74 | 0.31 |
| | Mid-to-deep | 9.3 | 0.86 | 0.38 |
| | Multi-block | 7.1 | 0.91 | 0.42 |
| XCiT-S12 | Early-to-mid | 13.8 | 0.71 | 0.26 |
| | Mid-to-deep | 11.2 | 0.81 | 0.35 |
| | Multi-block | 8.3 | 0.89 | 0.39 |

## B.9 DIFFERENT CLASSIFIERS & EXTENSION TO MULTI-CLASS CLASSIFICATION

PCG is not restricted to binary classifiers and extends in a straightforward way to multi-class settings by targeting the logit (or probability) of any chosen class. In the main text, we instantiate the explained classifier as a VGG-19 backbone fine-tuned for binary tasks to keep the exposition and visualisations focused.

In Figure 12, we illustrate this behaviour on AFHQ for a three-way classifier (cat / dog / wild) using two different backbones as the explained model: a ResNet-18 and a DenseNet-121. Each row starts from an input image (left) and shows PCG counterfactuals towards the two target classes classes for each backbone (e.g. cat → dog and cat → wild in the first row, dog → cat and dog → wild in the second, and wild → cat and wild → dog in the third). Across rows, PCG produces class-consistent transformations that primarily modify species-defining cues—such as ear and muzzle shape, fur texture, and overall facial structure—while preserving pose, lighting, and background. PCG is not tied to a specific classifier architecture: given a different multi-class backbone, the induced counterfactuals remain geometrically smooth and visually plausible, while reliably steering the prediction towards the desired target class.

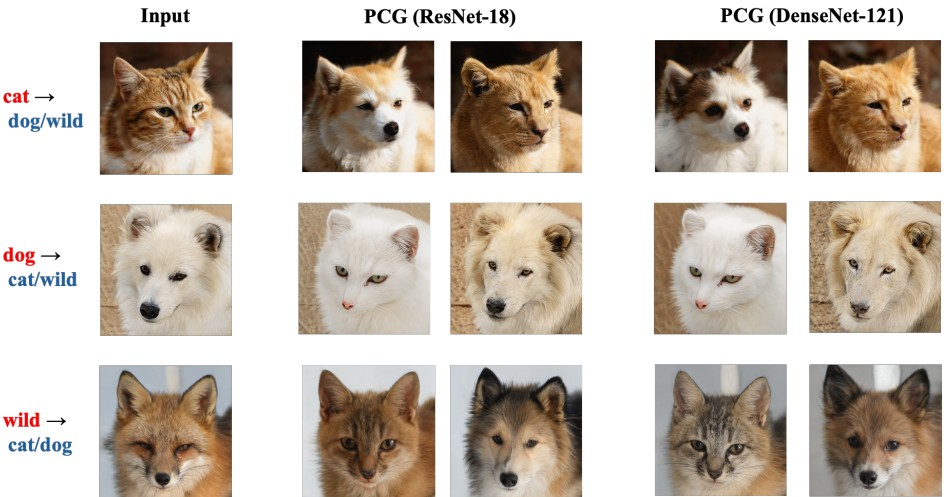

Figure 12: **PCG counterfactuals for multi-class AFHQ (cat / dog / wild).** From left to right: input images and PCG counterfactuals for ResNet-18 and DenseNet-121 classifiers, showing class-consistent cat↔dog↔wild transformations.

**Robust classifier with standard latent geometry vs. PCG**. In line with the literature on adversarial robustness, robust models are known to exhibit gradients and saliency maps that are more aligned with the data manifold and human perceptual structure than their standard counterparts. To probe how far this helps in the counterfactual setting, we consider a latent-space baseline without our robust perceptual metric, but with a robust classifier as the model to be explained. Concretely, we run VSGD in latent space using an AFHQ classifier fine-tuned from the same robust ResNet-50 that we use to induce our geometry, and compare it to PCG applied to a standard (non-robust) AFHQ ResNet-50 with our robust pullback metric.

Figure 13 shows qualitative examples on AFHQ. In all rows, PCG produces counterfactuals that are both class-consistent and tightly on-manifold: the species changes (ears, muzzle, fur pattern) while pose, lighting, and background remain stable, and there are no obvious local artifacts. VSGD with a robust classifier (right column) clearly improves over VSGD with a non-robust classifier—e.g., the cat→dog example yields a plausible dog face rather than a highly distorted image—reflecting the more manifold-aligned gradients of the robust model. However, the manifold conformity is still noticeably weaker than PCG. In the first row, the VSGD+robust counterfactual exhibits a larger semantic drift relative to the input than PCG: it achieves the target class, but with a more drastic change in identity and fine-scale structure, consistent with the absence of a geodesicity constraint. In the second and third rows, VSGD+robust produces dog/cat-like animals, but with local off-manifold artifacts in the fur and facial regions (e.g., irregular texture and shape in the mane/ears), whereas the PCG results remain visually smoother and more coherent. This aligns with our claim that a robust classifier guards against some adversarial behavior, but without an induced geometry that encodes its structure in latent space, trajectories can still slip into directions that are only weakly aligned with the data manifold.

| | **Input** | **PCG** | **VSGD + Robust Clf** |
| --- | --- | --- | --- |

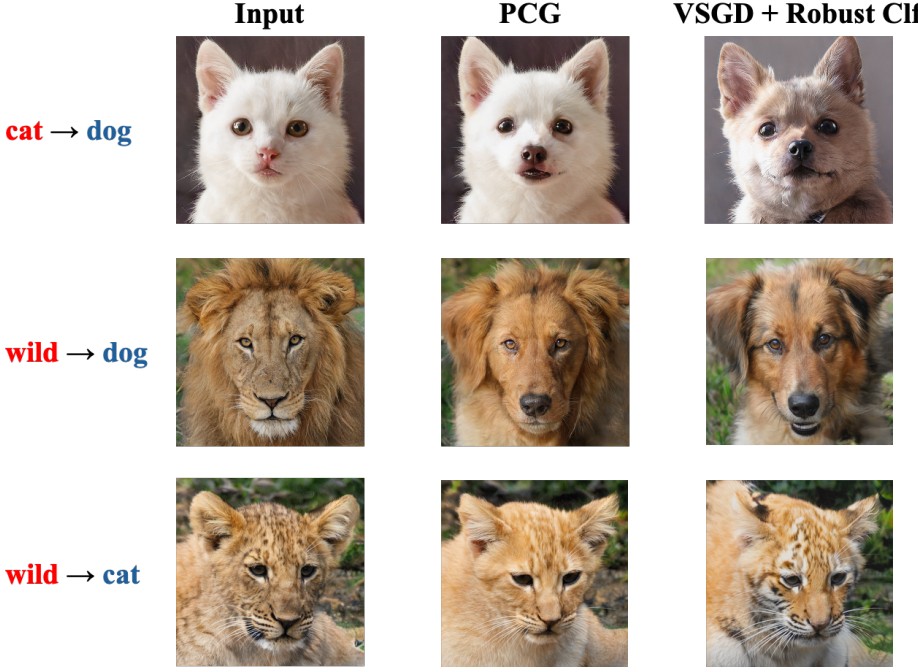

Figure 13: **PCG vs. VSGD with a robust classifier on AFHQ (cat / dog / wild).** Each row shows an input image (left), the PCG counterfactual for a standard ResNet-50 classifier (middle), and the counterfactual from VSGD applied to a robustly fine-tuned ResNet-50 classifier (right) for cat→dog and wild→dog/cat targets. VSGD with a robust classifier produces more meaningful counterfactuals than its non-robust counterpart but still exhibits larger semantic drift and local artifacts compared to PCG. For clarity, standard VSGD results with a non-robust classifier are omitted here, as they predominantly yield AEs rather than plausible counterfactuals.

To make this comparison systematic, we measure distributional realism (FID, R-FID), closeness (LPIPS, R-LPIPS), and manifold alignment under the robust geometry ($MAS_{rob}$):

Table 13: Comparison of PCG, VSGD, and VSGD with a robust classifier on AFHQ counterfactuals. Lower is better for FID, R-FID, LPIPS, and R-LPIPS; higher is better for $MAS_{rob}$ (STYLEGAN2 on AFHQ).

| Method | FID $\downarrow$ | R-FID $\downarrow$ | LPIPS $\downarrow$ | R-LPIPS $\downarrow$ | $MAS_{rob}$ $\uparrow$ |
| --- | --- | --- | --- | --- | --- |
| VSGD | 25.4 | 39.7 | 0.89 | 0.71 | 0.17 |
| VSGD + robust clf | 13.2 | 16.6 | 0.41 | 0.32 | 0.74 |
| PCG (ours, standard clf) | **8.5** | **9.4** | **0.23** | **0.19** | **0.89** |

Table 13 mirrors the qualitative picture. Switching from a standard to a robust classifier inside VSGD improves all metrics: FID and LPIPS decrease slightly, while R-FID, R-LPIPS, and $MAS_{rob}$ improve substantially, confirming that robust features reduce adversarial exploitation and encourage more semantic changes. Nevertheless, PCG remains clearly ahead, especially on the robust and geometry-aware metrics: its R-FID is close to the real target distribution, its R-LPIPS indicates smaller robust perceptual displacement from the original images, and its $MAS_{rob}$ is significantly higher, meaning that its counterfactual directions lie much more in the robust tangent space. In other words, VSGD+robust clf sits in an intermediate regime—better than non-robust latent optimization, but still less manifold-aligned than PCG. By contrast, PCG's behavior is driven primarily by the induced robust geometry: enforcing geodesicity forces the entire trajectory to conform tightly to the robust feature manifold, independently of whether the explanatory classifier is itself robust.

## C  FINAL NOTES ON SCOPE, LIMITATIONS & FUTURE WORK

Our work is deliberately centered on a specific but fundamental line of methods: latent-space instantiations of the Wachter et al. counterfactual objective, where one optimizes a counterfactual loss in the latent space of a generator and decodes back to image space. This family has been extremely influential, but, as we discuss in the introduction, it has accumulated chronic geometric failure modes in realistic vision settings (off-manifold traversals, on-manifold adversarial examples, and semantic drift along latent trajectories). These issues have pushed much of the field either to restrict latent-Wachter formulations to low-dimensional or tabular data, or to move to different paradigms altogether. PCG is best read as a geometric re-framing of this core latent-Wachter line: we make the choice of geometry explicit, induce it from robust perceptual features, and require trajectories to follow counterfactual geodesics rather than arbitrary latent updates. Our empirical focus and claims are therefore scoped to this regime: Wachter-style latent counterfactuals for non-robust classifiers in high-dimensional vision, under a fixed pretrained generator.

Within that scope, PCG targets the failure modes above and, under geometry- and robustness-aware diagnostics, improves both on realism, semantic proximity, and on manifold faithfulness relative to existing latent-space Wachter-based baselines. We do not claim to subsume all contemporary visual counterfactual methods. Other directions, discussed in our introduction, including pixel-space diffusion approaches (e.g. Weng et al., 2024; Sobieski et al., 2024) and latent diffusion methods (Sobieski & Biecek, 2024; Augustin et al., 2024; Luu et al., 2025) instantiate counterfactual generation through diffusion sampling, often with classifier guidance and regional constraints, in either pixel space or the latent space of a diffusion model. These are complementary directions: they work with different generative families and through optimization mechanisms different from the Wachter-style latent optimization.

Methodologically, PCG inherits several structural dependencies that can be viewed as limitations. First, like all latent-space approaches, PCG relies on a pretrained generator (and encoder, when present) whose latent space has reasonable manifold fidelity. Our robust pullback metric and two-stage path refinement mitigate artifacts from imperfect generators, but they cannot fully repair a severely mis-specified or collapsed latent space; in such cases, all latent methods, including PCG, are constrained by the quality of the underlying generative model. Second, the induced geometry is built from an adversarially trained robust backbone on ImageNet. This is motivated by theoretical and empirical work on adversarial robustness, which shows that robust models align gradients and feature variations more closely with the data manifold and human perceptual structure (Srinivas et al., 2024; Ganz et al., 2023a; Kaur et al., 2019a; Kettunen et al., 2019; Ghazanfari et al., 2024a). However, robustness is intrinsically local and distribution-dependent: the backbones used in our experiments are robust only within an $L_2$ ball of radius $\varepsilon$ around images drawn from (or close to) its training distribution. Our induced geometry is most trustworthy when (i) the generator and robust backbone operate on approximately the same domain, (ii) trajectories stay within regions where the robust model's predictions and gradients remain stable, and (iii) geodesic paths are reasonably well sampled with enough discretization of the underlying robust geodesic to allow the optimizer to distribute smooth semantic change across more intermediate states. Under strong distribution shifts, under-sampled paths, or strong displacements along trajectories that leave these neighborhoods, the semantic interpretation of the metric can degrade even though the construction remains mathematically valid.

These observations naturally point to future work. On the robustness side, it would be interesting to study how different robustness norms, radii, and training regimes (e.g., $L_\infty$ vs. $L_2$, certified vs. empirical robustness) reshape the induced geometry and the behavior of geodesics, and whether one can adapt the metric or the path optimization schedule to local estimates of robustness. On the generative side, our current choice of StyleGAN-family generators is partly pragmatic: they provide a single, well-defined latent space in which it is tractable to induce and analyze a metric via pullback geometry. Latent diffusion models, by contrast, involve multiple interacting spaces (the VAE latent, time-dependent diffusion states, and conditioning latents), and stochastic score dynamics over time. Transferring our geometric framework to that setting would require deciding where to place a geometry, how to define geodesics consistently along a stochastic trajectory, and how a pullback metric should interact with the score field. We expect the core phenomenon we study — that a strong generative prior alone does not preclude off-/on-manifold adversarial behavior when geometry is mis-specified — to persist in diffusion architectures, but a careful treatment is non-trivial and we view this as a distinct line of future work rather than a trivial extension.

Finally, our experiments are confined to images. Extending PCG beyond vision raises both modeling and geometric questions. Multimodal extensions could couple text and image spaces via joint latent geometries (e.g., CLIP-style or diffusion backbones) and cross-modal robust metrics; video counterfactuals would need to incorporate temporal coherence and spatiotemporal perceptual geometry; and applying similar ideas to graphs or language would require suitable generators and domain-specific robust features. In low-resource regimes, training full-scale robust backbones may be impractical, suggesting the need for lightweight robust surrogates, few-shot adaptation of perceptual metrics, or self-supervised proxies. We see PCG as a first step towards bringing explicit, robustly induced geometry into latent counterfactual explanations in vision, and anticipate that both the limitations and the structural ideas outlined here will be useful in guiding subsequent work across other generative families and data modalities.

