# OpenReview forum: "Counterfactual Explanations on Robust Perceptual Geodesics"
_ICLR.cc/2026/Conference — ICLR 2026 Poster_

### Official Review · Reviewer_T7oH · 2025-10-28

**Soundness:** 3
**Presentation:** 3
**Contribution:** 2
**Rating:** 4
**Confidence:** 3

**Summary:**

The paper studies the problem of generating counterfactual explanations for image classifiers.

The paper adopts a latent-space approach: the counterfactual explanation is found by traversing the latent space of a generative model.

The introduction clearly describes the problems faced by existing latent-space approaches and methods that search for counterfactual explanations more generally. For example, we don't want to provide on-manifold adversarial examples as counterfactuals.

The paper then proposes a novel method  to generate counterfactuals (PCG). The idea is to equip the latent space with a robust perceptual metric via pullback from a robust vision classifier (this is not the same classifier for which they generate the explanations).

The paper states that when counterfactual explanations are generated in the latent space using the robust perceptual metric, they correspond to smooth, on-manifold, semantically valid transitions between images of different classes.

**Experiments:** The novel method PCG is compared against other latent-space counterfactual methods. The paper contains the following evaluations:

- Visual inspection of interpolation paths in the latent space (Figure 2, Figure 3)
- Quantitative comparison between different CF generation methods according to standard metrics (Table 1)
- Qualtitative dedication of counterfactuals generated by different methods (Figure 4).

**Strengths:**

- The proposed method is well-motivated and extensively described in the paper
- The contribution of the paper is situated within the existing literature
- The empirical results in the paper are encouraging, both for the geodesics and the generated counterfactual explanations
- The paper is clearly written

**Weaknesses:**

**Limited empirical results:** While the motivation and description of the proposed method are extensive, the breadth of the empirical results is surprisingly limited. By this I mean that:

- The qualitative examples in the paper are almost all either for humans or for cats and dogs. So there is a limited breadth in terms of the classes for which the counterfactuals are explored.
- The classifiers for which the counterfactuals are generated are VGG-19 backbones trained on a binary classification task. This seems like a rather specific choice; it should be relatively easy to extend this to a multi-class setup with other architectures?
- Some details, like class probabilities assigned by the classifiers, are not reported

For example, I compared the empirical results in this paper to the results in "Diffusion Visual Counterfactual Explanations" NeurIPS 2022 (https://proceedings.neurips.cc/paper_files/paper/2022/hash/025f7165a452e7d0b57f1397fed3b0fd-Abstract-Conference.html). This paper does not use a latent-space approach, but, similarly to the submitted work, it proposes a method to generate robust counterfactual explanations for arbitrary classifiers. This paper has similar limitations to the submitted paper, insofar as it must rely on a visual inspection of the generated counterfactuals and some proxy metrics. However, the empirical results in this paper are much more detailed.

**Unclear if the proposed approach beats the state of the art:** While the proposed method is very well-motivated, the limited empirical evaluation does not make clear to me in what precise sense the obtained counterfactuals can be considered state of the art among latent space methods, or method for generating counterfactual explanations for non-robust classifiers more generally (if we were to compare with diffusion-based approaches as well, as in the above reference)

**Questions:**

**Question 1:** Why is the proposed approach not applied to arbitrary ImageNet classifiers? (or other standard classifiers from PyTorch repositories)

**Question 2:** If we were to generate counterfactual explanations using a latent-space approach without your robust perceptual metric, but for a robust computer vision classifier (like the robust model that you derive your metric from), would this result in similar explanations?

**Question 3:** Is the direction from the original image to the generated counterfactual approximately aligned with the tangent space of the data manifold at the original image, or is this not the case?

**Reason for Final Score:**  The strengths of this paper is a very well-written introduction and a very good motivation for the proposed method, including the mathematical details. The main weakness of this paper is the limited scope of the empirical results, which fall somewhat below the bar for a conference like ICLR. To me, this makes the current version of the paper a marginal paper. I decided to assign the paper a score of 4 for now.

---

> ### Author Response · Authors · 2025-11-21
>
> Responses to Weaknesses & Questions
>
> **On the breadth of classes and tasks.**
> The qualitative examples in the main text and appendix may give the impression of a narrow focus, but we in fact run our experiments on a wider set of class pairs across all three datasets, and show at least a few examples for each pair. On AFHQ, we consider *cat vs dog*, *wild vs dog*, and *cat vs wild*, where *wild* spans multiple species (e.g., fox, cheetah, lion). On FFHQ, we train separate binary classifiers on CelebA for *non-smile vs smile*, *bald vs non-bald*, *blonde vs non-blonde*, *with glasses vs without glasses*, and *male vs female*. We initially trained a larger pool of attribute classifiers and retained this subset because they achieved reliable validation performance; other attributes suffered from the usual CelebA issues (e.g., class imbalance, label noise, etc.). On PlantVillage, we use *healthy vs non-healthy* leaves, with the *non-healthy* class aggregating two distinct disease categories.
>
> The quantitative results in the main text and in relevant appendices are computed across all of these tasks for the specified datasets, not only the *cat/dog* or *face* examples shown as part of the main figures.
>
>
> **On weakness 2, 3, and question 1: Beyond VGG-19 backbone classifiers & multi-class tasks.**
> PCG indeed is not restricted to binary classifiers, and extends in a straightforward way from binary to multi-class by targeting a desired logit for any chosen class. In the original submission, we chose a VGG-19 backbone fine-tuned for a binary task mainly to keep the exposition and visualisations focused, not because PCG is restricted to that setting. The only structural requirement our method imposes is that the to-be-explained classifier operates on the same data domain as the generator—i.e., it predicts over the classes for which the generator has learned a meaningful latent representation. In that sense, PCG is compatible with arbitrary classifiers as long as they are trained on (or close to) the generator’s distribution; all latent-space counterfactual methods inherit this coupling to the underlying generator.
>
> In the revised manuscript, we make this explicit and go beyond the initial binary VGG-19 example. In **Appendix B.9**, we apply PCG to multi-class classification on AFHQ and instantiate the explained model as standard ImageNet-style architectures, including DenseNet-121 and ResNet-18. Across these settings, we observe the same qualitative behaviour and geometric advantages, which illustrates that the approach is not tied to a specific backbone or to binary classification, but extends naturally to multi-class tasks and other standard classifiers.
>
>   - **Class probabilities.**
> Thank you for pointing this out. We plan to add the classifier’s predicted probabilities above each image in the next revised submission to make class-level attribution more explicit.
>
> **On scope: Latent Wachter-style methods**. Our work is deliberately centred on a specific, but fundamental, line of methods: latent-space instantiations of the Wachter et al. counterfactual objective, where one optimizes a counterfactual loss in the latent space of a generator and decodes back to image space. This direction has been extremely influential, but, as we detailed in the introduction, it has accumulated a set of chronic geometric failure modes in vision (e.g., off-manifold traversals, on-manifold AEs, and semantic drift along latent trajectories) which have led much of the community either to constrain these methods to low-dimensional or tabular settings, or to move away from the Wachter formulation entirely toward alternative paradigms, such as methods based on diffusion counterfactuals that generate explanations via different mechanisms.
>
> Our goal in this paper is to revive and regularize this latent-Wachter line by placing it on a principled geometric footing that explicitly accounts for the inevitable distortions introduced by deep generators: we make the choice of induced geometry explicit (via a robust pullback metric), and we require all updates and paths to conform to this geometry through counterfactual geodesics, rather than arbitrary latent traversals. This framework systematically targets the long-standing geometric failure modes of latent methods in high-dimensional vision. Diffusion-based approaches represent an important, complementary departure from Wachter-style latent optimisation, and/or they operate under different assumptions and mechanisms. PCG is thus best viewed within the direction we specified as a geometric re-framing of a core, widely used counterfactual objective that has struggled to scale reliably to realistic vision domains. In the current revision, we have situated the diffusion-based direction along the same axes used throughout our paper in order to clarify their relationship to our setting (see **Appendix C**).

---

> > ### Author Response · Authors · 2025-11-21
> >
> > **On Q3: Manifold alignment**. We thank the reviewer for this question. To address it, we introduce a *Manifold Alignment Score (MAS)* that measures, for each method, how much of the change from the original image to its counterfactual lies *along* the data manifold (tangent directions) versus *orthogonal* to it (off-manifold directions), under different geometries (pixels, standard features, robust features). Concretely, for each geometry, we
> > (i) map both the original and counterfactual images into the corresponding feature space,
> > (ii) consider the vector that connects them in that space, and
> > (iii) decompose this vector into a tangent component (estimated from the generator plus the chosen geometry) and a normal component.
> >
> > MAS is defined as the fraction of the normalized movement that stays in the tangent space; scores lie in $[0,1]$, with higher values indicating better alignment with the manifold at the original point. The formal definition and implementation details are provided in **Appendix A.5**.
> >
> > **Behaviour across methods.**
> > As reported in **Table 2, Page 10**
> >
> > - Methods that do *not* explicitly model geometry (REVISE, VSGD) exhibit low MAS under all geometries. Their counterfactual directions have a large normal component—i.e., they tend to move off the manifold regardless of whether we measure in pixel, standard-feature, or robust-feature space.
> > - RSGD shows high alignment in pixel space—the geometry it directly optimizes under—but poor alignment in standard and robust feature geometries. This indicates that while RSGD follows the pixel-geometry manifold, it still strays from what standard or robust semantic features consider to be manifold-consistent directions.
> > - RSGD-C improves alignment under the standard feature geometry it implicitly optimizes (higher MAS in that column), but shows substantially lower alignment under the robust geometry. Its directions are tangent for the standard classifier features, but not for robust perceptual features.
> > - PCG achieves the highest MAS under the robust geometry (as expected from its construction), and also maintains highest alignment under the standard feature geometry and reasonable alignment in pixel space. In other words, the direction from the original to the PCG counterfactual is well-aligned with the tangent structure of the robust feature manifold at the original point, and this alignment transfers to standard features as well.
> >
> > **On Q2: Robust classifier + standard latent geometry vs PCG.**
> > For this, we consider a latent-space approach without our robust perceptual metric, but with a robust classifier as the one-to-be-explained. Concretely, we take VSGD in the latent space and optimize counterfactuals for an AFHQ classifier robustly fine-tuned from the robust ResNet-50 model we used. In parallel, we run PCG with a standard ResNet-50 (non-robust) AFHQ classifier, using the robust pullback geometry and geodesic optimisation.
> >
> > Robust models have been shown to have gradients and saliency more aligned with the data manifold and perceptual structure than standard models [1-4], and in line with this literature we indeed find that VSGD + robust classifier guards against adversarial changes and produces more meaningful counterfactuals than VSGD with a non-robust classifier. However, qualitatively, these counterfactuals still exhibit local artefacts and semantic drifts that indicate a relatively weaker conformity to the manifold than what we observe with PCG. We illustrate this in the revision in **Appendix B.9.**
> >
> > To make this comparison systematic, we also measure **FID**, **R-FID**, **LPIPS**, **R-LPIPS**, and the **manifold alignment score** under the robust geometry (see **Appendix B.9.**  for results and **Appendix A.5** for details on the evaluation metrics).
> >
> > We find that robustness of the classifier does help, but does not fully resolve the geometric pathologies we discussed of latent optimisation. VSGD with a robust classifier sits in an intermediate regime: better than non-robust latent baselines, but still noticeably less aligned with the manifold than PCG.
> >
> > By contrast, PCG’s behaviour is driven by the geometry: inducing a robust pullback metric and optimising a geodesic energy forces the trajectory itself to conform tightly to the robust feature manifold, independent of whether the explanatory classifier is robust. In this sense, robust models make meaningful counterfactuals more likely, but without a geometry that encodes their structure in latent space, trajectories can still slip into directions that are only weakly manifold-aligned. Our experiment makes this distinction explicit.
> >
> >
> > [1] Srinivas et al. Which Models have Perceptually‑Aligned Gradients? An Explanation via Off‑Manifold Robustness
> > [2] Ganz et al. Do Perceptually Aligned Gradients Imply Adversarial Robustness?
> > [3] Kaur et al. Are Perceptually-Aligned Gradients a General Property of Robust Classifiers?

---

> ### Author Response · Authors · 2025-11-21
>
> **Overall: More quantitative evaluations.**
> We appreciate the request for more empirical evaluation. In the revised manuscript, we include extensive quantitative metrics and ablations across tasks, architectures, and geometries:
>
> - **Distributional Realism:**
>   *FID*, *R-FID*
>
> - **Perceptual Smoothness:**
>   *LPIPS*, *R-LPIPS* (for point-wise similarity and path-based smoothness)
>
> - **Semantic Faithfulness:**
>   *Semantic Margin (SM)*, *COUT* (classifier output sparsity)
>
> - **Manifold Consistency:**
>   *Manifold Alignment Score (MAS)*
>
> - **Stability under Random Seeds:**
>   *CF Dispersion*, *Contraction Ratio*, *CF Diameter*
>
> We also report ablations on:
> - Endpoint loss weight $\lambda$ underlying PCG
> - Robust backbone type (ResNet vs ViT)
> - Layer aggregation strategies
> - Robust classifier vs robust geometry
> - Extensions to different backbones and multiclassification

---

> > ### Comment · Reviewer_T7oH · 2025-11-21
> > **Respone to author rebuttal**
> >
> > Thank you for the very detailed rebuttal. I appreciate that. Based on your response and the additional promised changes, I have increased my score. I'm probably the reviewer who knows this paper's topic least, so my final assessment might be based on discussion with the other reviewers.

---

> > > ### Author Response · Authors · 2025-11-21
> > >
> > > Thank you for updating your score. We appreciate the time you invested in the review and your engagement with the rebuttal.

---

### Official Review · Reviewer_RJrE · 2025-10-28

**Soundness:** 2
**Presentation:** 3
**Contribution:** 3
**Rating:** 4
**Confidence:** 4

**Summary:**

This paper addresses an important problem of the potential adversarial nature of counterfactual explanations (CEs) in computer vision. To avoid off- and on-manifold adversarial examples (AEs) as solutions to the empirical CE optimization problem, the authors propose to perform the optimization within the latent space of a generative model with the geometry induced by the aggregated representations of a robustly trained classification model. This optimization takes into account the entire path traced from the true sample to the CE and aims at finding a geodesic between these points wrt. the proposed geometry. Within the experimental section, the authors aim at showing improved smoothness in comparison to baselines, superior results in terms of several geometric measures and the ability to avoid off- and on-manifolds AEs.

**Strengths:**

S1. The considered problem is often overlooked in the CE context, especially in the computer vision domain, where human perception may be easily fooled. It is also of extreme importance to the explainability community -- since CEs stand at the top of the Pearl's causality ladder, it is crucial to generate explanations that are truly valid. The authors provide an elegant solution to this problem, with clearly highlighted motivation and proper mathematical formalism.

S2. The proposed PCG algorithm is actually a novel, two-phase approach with clear motivation behind its construction, especially for phase I. The clarity of the description is great, the baselines are properly described, and the overall flow of the paper is enjoyable for the reader.

S3. Prior methods in the undertaken research direction were mainly considered in the tabular setting. The authors did a very good job in transferring and adapting these methods to the vision domain.

**Weaknesses:**

W1. Robust models are never "infinitely" robust (lines 225-227), meaning that their robustness is preserved up to some nieghborhood of each point. What are the ways of measuring this robustness? How does the level of this robustness influence the resulting induced geometry? What limitations spark from that and how can they be overcome?

W2. Is there some theoretical justification for the definition of the robust perceptual metric $G_R$ (its equation is not numbered, but can be seen at line 234)? Or is it purely a heuristic choice? I fully understand the motivation behind it, but this particular choice of the definiton appears in a slightly surprising way without any reference, especially a theoretical one about the robustness itself.

W3. Influence of $\lambda$ seems crucial as it may significantly impact the robust energy and eventually make the trajectory misaligned with the induced robust semantic geometry. The paper currently lacks any ablation study related to $\lambda$'s influence.

W4. I am fully aware that evaluating the claimed improved smoothness and manifold consistency is very challenging. However, I must argue that simply providing a few example trajecteries is not enough to properly back the claims. One way of addressing this is through a user study, in which participants are asked about the perceived smoothness, i.e., the ground-truth for the semantic similarity perception. The other, probably less demanding and more suitable to the rebuttal, is finding some proxy measure to quantitatively assess the properties.

W5. While I agree that counterfactuals from other methods seem to be off- or on-manifold AEs, how can this be measured at scale? What is the guarantee that these samples are not just cherry-picked? A quantitative comparison here is important to assess whether PCG is not suffering from the same problems.

W6. The paper currently does not contain any explanandum-related metrics and is missing the typical ones from prior work. While I do understand that baseline works did not use them, the extension to vision domain deems it necessary (at least for me). The paper should additionally contain a comparison in terms of FID ([1], closeness at the distribution level), COUT ([2], sparsity relative to the explained model) and flip/sucess rate (effectiveness of the method).

W7. Important ablations are missing: what is the influence of the choice of the robust model? how does its training data influence the resulting geometry (the proposed method relies on ImageNet-based model everywhere)? How does the choice of the number and weights of the layers affect it?

W8. The paper is currently ignoring two important research directions. First one is related to current state-of-the-art solutions based on pixel-space diffusion-based models, e.g. [3] and [5]. These should be at least mentioned as recent methods that do not focus on the problem considered in the paper, which will only strengthen its relevance. The other direction considers latent-diffusion-based methods, which also perform some kind of optimization in the latent space of a VAE, but with an addition of a diffusion model [4,6,7]. What is the influence of the VAE+diffusion combination in the considered problem of off- and on-manifold AEs? These issues should be at least mentioned in the paper to ensure maximum impact.

Minor:
- line 095: the mentioned methods are mostly in the vision domain, except the first one
- line 238: neural networks (for prediction tasks) typically compress data, so it should probably be $d_k << D$
- line 921: is the change to Softplus actually relevant? If yes, some experiment should be showing that in the paper. If not, why not just remove this procedure and stick with ReLU?
- the main text should be slightly more precise about phase II, the re-anchoring procedure and the overall details. It is fully understandable only after referring to the pseudocode from the appendix.

[1] Heusel et al., GANs Trained by a Two Time-Scale Update Rule Converge to a Local Nash Equilibrium, NeurIPS, 2017

[2] Khorram et al., Cycle-Consistent Counterfactuals by Latent Transformations, CVPR, 2022

[3] Weng et al., Fast Diffusion-Based Counterfactuals for Shortcut Removal and Generation, ECCV, 2024

[4] Sobieski and Biecek, Global Counterfactual Directions, ECCV, 2024

[5] Sobieski et al., Rethinking Visual Counterfactual Explanations Through Region Constraint, ICLR, 2025

[6] Augustin et al., DiG-IN: Diffusion Guidance for Investigating Networks - Uncovering Classifier Differences Neuron Visualisations and Visual Counterfactual Explanations, CVPR, 2024

[7] Luu et al., From Visual Explanations to Counterfactual Explanations with Latent Diffusion, WACV, 2025

**Questions:**

Overall, I am a big fan of the proposed approach and consider the problem to be extremely relevant. My greatest concerns lie in the scope and scale of the current evaluation scheme, which does not provide sufficient evidence for some of the claims and does not shed light on important aspects of the proposed method, e.g, the $\lambda$ hyperparameter. During the rebuttal, please refer to the weaknesses above, which either contain the questions directly or implicitly.

---

> ### Author Response · Authors · 2025-11-21
>
> We thank the reviewer for their careful reading and serious feedback!
>
> **On W1 and W2: Robustness, induced geometry, and the robust perceptual metric.**
> W1 and W2 both touch on the robustness assumptions behind our construction. We address them together here, restating and extending our answer to Reviewer Bs9x on a similar point.
>
> **Grounding.** First, our use of robust features is motivated by concrete theoretical and empirical results in adversarial robustness and *Perceptually Aligned Gradients* (PAGs). [1] show that adversarially trained models exhibit saliency maps that are more strongly aligned with human salient structure and suppress high-frequency, non-salient directions. [2] give a theoretical account via *off-manifold robustness*, showing that if a classifier is trained to be more robust off the data manifold than on it, its input gradients are forced to lie approximately in the tangent bundle of the data manifold. This provides a mechanism for why robust models tend to have gradients that follow intrinsic manifold directions rather than adversarial spikes orthogonal to the data. This picture is consistent with the broader PAG literature, where works such as [3] and [4] study how robust models exhibit gradients that align with human perceptual judgments.
>
> **Human perceptual alignment via robustness.** Second, this manifold alignment carries over to feature-based perceptual metrics. The literature on robust perceptual similarity shows that distances in robust feature spaces correlate better with human similarity judgments and are less vulnerable to adversarial manipulation than their standard counterparts. [5] demonstrate that LPIPS-type metrics are themselves vulnerable and can be rectified into a robust variant by using robust representations. [6] go further and construct a 1-Lipschitz perceptual similarity metric with provable robustness guarantees, and show that robustness in the feature space improves both adversarial stability and alignment with human perceptual similarity. Together with the manifold alignment results above, these works support the view that infinitesimal moves in robust feature space correspond to smooth, semantically coherent deformations along the data manifold.
>
> **Our metric.** Our robust pullback metric is designed to encode exactly this structure. We aggregate intermediate representations (to capture semantics at different levels of abstraction) from an adversarially trained ImageNet backbone (L2-robust ResNet-50 with $\varepsilon=3.0$) and pull them back through the generator. In differential geometric terms, this is the standard pullback construction: a Riemannian metric on the feature space induces a Riemannian metric on the latent space via the generator map. The aggregation is chosen to encode feature abstractions at different level of granularity. Geodesics under this metric are therefore biased to track the manifold-aligned, perceptually meaningful directions that robust models use internally, rather than the brittle directions that standard Euclidean pixel space or fragile feature spaces often follow. In this sense, the metric is a natural consequence of combining standard pullback geometry with robust feature spaces that are known to be better aligned with both the data manifold and human perception.
>
>
> **Locality and guarantees.** Turning specifically to W1, we fully agree that robustness is intrinsically local and depends on the choice of norm and radius used in training. The backbone we use is adversarially trained for $\ell_2$-bounded perturbations with radius $\varepsilon = 3.0$ on ImageNet. Conceptually, this defines a local robustness region around each point in image space: within roughly an $\ell_2$ ball of radius $\varepsilon$, the classifier is empirically stable and its gradients and features exhibit the alignment properties documented in the aforementioned works. Our induced geometry should therefore be understood as most semantically reliable within such neighbourhoods. If a geodesic or counterfactual path remains within regions where the robust model preserves its prediction and its gradients remain well behaved, then the pullback metric reflects the intended robust perceptual geometry. If a path were to traverse far beyond those regions (e.g., under strong distribution shift), the robustness guarantees weaken and the induced geometry may no longer coincide with human semantic structure, even though it remains mathematically well defined.

---

> > ### Author Response · Authors · 2025-11-21
> >
> > **On W1 and W2: Continued.**
> >
> > This locality introduces two practical limitations. First, robustness is defined with respect to the training distribution of the robust model. This is why we deliberately choose a robust backbone trained on ImageNet and restrict our datasets to lie close to the ImageNet domain, so that the robust features and gradients are operating in the regime in which they were trained and validated. Second, robustness is defined with respect to a specific norm and radius. An $\ell_2$ $\varepsilon$ of 3.0 produces a certain trade-off between invariance and sensitivity. Under larger radii, the robust model would enforce stronger invariances and potentially larger trust regions, at the cost of accuracy; under smaller radii, the local trust region shrinks. Exploring how different norms and radii (and how that interplay affects the choice of Riemannian metrics in the chosen ambient space) change the shape of the induced geometry and the behaviour of geodesics is a natural direction for future work, but is beyond the scope of the current paper.
> >
> > **When robustness alone is insufficient**. At the same time, we do not present this as a universal guarantee. The induced geometry can degrade when there is a strong distribution shift between the robust model’s training domain and the generator’s domain. Similarly, if the generator does not faithfully parameterize the data manifold (e.g., known failure modes in GANs such as mode collapse; [7]), no metric can fully prevent semantically problematic traversals. Our induced geometry is most trustworthy when (i) the generator and robust backbone operate on approximately the same domain, and (ii) geodesic paths are reasonably well sampled with enough discretization of the underlying robust geodesic to allow the optimizer to distribute smooth semantic change across more intermediate states (see **Appendix B.5** on how the number of points on the path can affect smoothness).
> >
> > Under strong distribution shifts, under-sampled paths, or strong displacements along trajectories that leave these neighborhoods, the semantic interpretation of the metric can degrade even though the construction remains mathematically valid. We have now included this discussion in the revision.
> >
> > Under these conditions, inducing geometry from robust features provides a principled mechanism for biasing counterfactual geodesics toward semantically meaningful paths.
> >
> > Finally, to address the concern that the behaviour of the induced geometry might be tied to a specific robust model or layer choice, we have added ablations over both the robust backbone and the feature aggregation scheme (see our reply to W7). In particular, we replace the robust ResNet-50 with a robust Vision Transformer and vary which layers or blocks are used to build the metric, and we observe that the behaviour and advantages of PCG remain stable.
> >
> > ---
> >
> > **References**
> > [1] Etmann et al. *On the Connection Between Adversarial Robustness and Saliency Map Interpretability*
> > [2] Srinivas et al. *Which Models Have Perceptually‑Aligned Gradients? An Explanation via Off‑Manifold Robustness*
> > [3] Ganz et al. *Do Perceptually Aligned Gradients Imply Adversarial Robustness?*
> > [4] Kaur et al. *Are Perceptually-Aligned Gradients a General Property of Robust Classifiers?*
> > [5] Kettunen et al. *E-LPIPS: Robust Perceptual Image Similarity via Random Transformation Ensembles*
> > [6] Ghazanfari et al. *LipSim: A Provably Robust Perceptual Similarity Metric*
> > [7] Karras et al. *Analyzing and Improving the Image Quality of StyleGAN*

---

> ### Author Response · Authors · 2025-11-21
>
> **On W3: Ablation effects of $\lambda$ on PCG.**  We added an explicit ablation on the endpoint loss weight $\lambda$. We compare static and scheduled values of $\lambda$ over a fixed optimization budget and report:
> (i) a **target-retention rate** (fraction of runs where the endpoint remains in the target class under a specified budget), and
> (ii) **FID / Robust FID (R-FID)**, **LPIPS / Robust LPIPS (R-LPIPS)** (see **Appendices A.5, B.6**).
>
> ---
>
> **On W4: Quantitative assessment of counterfactual geodesic smoothness.**  To address this, we use path-based proxy metrics centred on the PCG counterfactual geodesic and compare them to standard interpolation paths between the same endpoints. For each input, we consider three trajectories with the same number of points between $x_{\mathrm{orig}}$ and $x_{\mathrm{cf}}$:
> (i) the **PCG path** produced by our two-phase optimization in the induced robust geometry,
> (ii) a **linear interpolation** in latent space, and
> (iii) a **spherical (slerp) interpolation** in latent space.
>
> For any such path, we measure the average perceptual change between consecutive frames using $\overline{\Delta \mathrm{LPIPS}} $, and we also report a robust counterpart (R-LPIPS) where the backbone is replaced by a robust model better aligned with human perceptual structure. Intuitively, smaller average steps in these metrics correspond to smoother, more gradual changes along the trajectory, with R-LPIPS putting more weight on semantically meaningful variation rather than non-robust directions.
>
> **Result.**
> PCG geodesic achieves the smallest average step size under both LPIPS and R-LPIPS, while latent linear and spherical baselines exhibit larger and more irregular step sizes. This behaviour is consistent with the construction of PCG itself: in the continuous setting, a geodesic in a Riemannian metric is the curve that minimises path energy (and hence length) and moves at constant speed, playing the same role as straight lines in Euclidean space. Our discrete optimization approximates such a geodesic in the induced robust perceptual geometry, so the PCG path is, by design, the shortest and smoothest “perceptual” connection between $x_{\mathrm{orig}}$ and $x_{\mathrm{cf}}$ under that metric (**Appendix B.7**).
>
> The empirical gap in average (R-)LPIPS provides a quantitative counterpart to this geometric picture. Manifold consistency/alignment (in the sense of how closely the counterfactual displacement follows the data-manifold tangent under each geometry) is evaluated separately via the tangent-alignment score introduced in our response to W5.
>
> ---
>
> **On W5: More systematic and scalable quantification of baseline failure modes.**
> We appreciate the reviewer’s request for more quantitative evaluation on whether the reported issues with baseline methods hold systematically. In the revision, we address this point via two quantitative metrics applied at scale that probe:
> (i) whether counterfactuals move into regions genuinely associated with the target class, and
> (ii) whether the direction of change is aligned with the tangent spaces induced by different geometries.
>
> Both metrics are applied to all generated counterfactuals for each method (see **Pages 9–10**, and **Appendix A.5** for definitions).
> - The first is a *semantic margin* in an independent robust feature space: for each counterfactual, we compare how close it sits to target-class training examples versus non-target examples. Positive margins indicate that the counterfactual has moved into a region of feature space genuinely populated by target-class data; non-positive margins flag samples that remain in mixed or non-target neighbourhoods and are thus suspect from a manifold perspective.
> - The second is a *manifold alignment score* that measures how much of the counterfactual displacement lies in the tangent spaces induced by different geometries (pixel, standard features, robust features), capturing whether the change is predominantly tangent (manifold-aligned) or has a large normal component (off-manifold or geometry-misaligned).
>
> **Result.**
> Empirically, PCG achieves the largest positive semantic margins and the highest tangent-alignment scores under both the robust and standard feature geometries. Methods that do not model geometry exhibit substantially lower scores. Taken together, these two metrics provide a precise, scalable characterisation of the failure modes we highlight: PCG counterfactuals not only cross the decision boundary, but move into target-populated regions and follow the induced manifold structure, whereas baselines often fail on one or both axes.

---

> > ### Author Response · Authors · 2025-11-21
> >
> > **On W6: Explanandum-related metrics (FID, R-FID, LPIPS, R-LPIPS, COUT, flip/success rate).** Our main evaluation focuses on geometry- and robustness-aware behaviour, but we agree that distributional and explanandum-style metrics are useful for situating PCG within the counterfactual literature. We therefore report both the standard FID and a robust variant R-FID, LPIPS, and its robust counterpart R-LPIPS, together with a motivated COUT sparsity metric (see **Pages 9–10**, and **Appendix A.5**). We intentionally do not use flip/success rate as a quality signal for reasons discussed below.
> >
> > **On the robustness of FID.**
> > FID measures how close the distribution of generated images is to that of real images in a standard feature space. It is widely used, but recent work [1] has shown that standard FID can be surprisingly vulnerable: models can improve FID via imperceptible or non-semantic perturbations, and FID can fail to track perceptual or semantic fidelity when the feature extractor is not robust. LPIPS is also vulnerable in this adversarial regime, for which we use R-LPIPS [2]  This is precisely the kind of exploitability we highlight in the paper.
> >
> > **Robust FID.**
> > To address this, we additionally report R-FID, where the feature extractor is replaced by a robust counterpart. The form of the metric is the same, but it is computed in a feature space that is adversarially robust and better aligned with perceptual and semantic structure. Intuitively, if a method exploits non-robust directions, its standard FID may still look reasonable, while its R-FID will deteriorate.
> >
> > **Empirical result.**
> > PCG performs best under standard FID, but the gaps becomes significantly larger under R-FID: PCG remains close to the real target distribution in robust feature space and induces more targeted changes in the classifier’s representation (as measured by COUT), while baselines degrade more strongly and rely on less structured internal shifts. This is consistent with the broader message of the paper: in adversarially vulnerable vision settings, standard distributional metrics alone can be overly optimistic and may not distinguish genuinely semantic counterfactuals from adversarial artefacts, whereas robustified metrics and geometry-aware diagnostics reveal the advantage of a semantics-aware, robustly induced geometry.
> >
> > We intentionally do not **flip/success rate** as it almost perfectly obscures the failure mode we are trying to call out. Both a robust, semantically meaningful CEs and a brittle AE are defined by the same condition: they both succeed in flipping the classifier to the target class. In our experiments, all methods achieve similarly high flip rates, including those that visibly produce on-/off-manifold adversarial artefacts under our geometric and robust evaluations. This confirms the concern we raise: metrics that only check “does the label flip?” can be optimized by adversarial behaviour and do not reflect whether the underlying mechanism is explanatory.
> >
> > We highlight that under standard FID, PCG is best with a larger gap under R-FID, showing that it  remains close to the real target distribution in a robust feature space, while baselines degrade much more and that non-robust, distribution-level metrics can be partially satisfied even by methods that rely on adversarial artefacts. This is aligned with [3] that showed that adversarial perturbations are highly predictive, generalizable features under the data distribution. PCG is also best under LPIPS and R-LPIPS. COUT follows the same pattern, with PCG inducing more targeted changes in the classifier’s representation than baseline methods.
> >
> > ---
> >
> > **References**
> > [1] Alfarra et al. *On the Robustness of Quality Measures for GANs*
> > [2] Kettunen et al. *R-LPIPS: An Adversarially Robust Perceptual Similarity Metric*
> > [3] Ilyas et al. *Adversarial Examples Are Not Bugs, They Are Features*

---

> > > ### Author Response · Authors · 2025-11-21
> > >
> > > **On W7: Influence of the choice of robust model and layer aggregation.** We agree that the induced geometry depends on the robust backbone and on how its internal layers are aggregated. Our main experiments use a pretrained $\ell_2$--robust ResNet-50 on ImageNet to define the perceptual metric. We have now added ablations along two axes: (i) switching to a robust Vision Transformer–style backbone, and (ii) varying the layer sets used to induce the metric for both backbones.
> > >
> > > **Backbone choice.**
> > > In addition to the robust ResNet-50, we induce geometry from an $\ell_2$--robust XCiT-S12 [1] (ViT-style) backbone trained on ImageNet under the same threat model. In all cases, the generator and the explained classifier $f$ are kept fixed; only the feature map used to define the metric changes.
> > >
> > > **Layer aggregation.**
> > > For each backbone, we consider three configurations:
> > > (i) *Early-to-mid:* early and intermediate layers/blocks (ResNet-50: stem + conv2\_x; XCiT-S12: blocks 3 and 5).
> > > (ii) *Mid-to-deep:* later, more semantic layers/blocks (ResNet-50: conv3\_x--conv5\_x; XCiT-S12: blocks 7, 9, 11).
> > > (iii) *Multi-block (default):* aggregation across early-to-deep layers.
> > >
> > > For each configuration, we recompute robust FID (R-FID), the manifold-alignment score under the robust geometry (MAS$_\text{rob}$), and the robust feature-space semantic margin (mean SM).
> > >
> > > **Findings.**
> > > (i) Across architectures (CNN vs. ViT-style), PCG consistently outperforms all baselines; absolute values shift slightly between backbones, but the relative trends and performance gaps remain stable.
> > > (ii) Within each backbone, layer ablations show a clear pattern: using only early-to-mid layers yields weaker tangent alignment and semantic locality; using only mid-to-deep layers improves both substantially; and aggregating early + mid + deep layers (*multi-block*) gives the strongest overall performance in R-FID, MAS$_\text{rob}$, and mean SM.
> > >
> > > These results are now included in **Appendix B.8**.
> > >
> > > **On W8: Other relevant research directions.**  We thank the reviewer for highlighting these important directions. We agree that they represent a valuable and growing line of work, and we now explicitly discuss and cite them in the revised manuscript (see **Appendix C**). In particular, we have situated these works along the same axes used throughout our paper in order to clarify their relationship to our setting. While these approaches differ methodologically—especially in how they formulate counterfactual generation, and utilize different optimization mechanisms, we believe they meaningfully complement our work. We now clarify how they relate to, and complement, the Wachter-style latent counterfactual setting we focus on in this paper.
> > >
> > > - **Latent Diffusion Geometry**. Regarding VAE plus diffusion combinations and their influence on off- and on-manifold AEs, our expectation is that the core issue we study would carry over: a stronger generative prior improves realism but does not on its own guarantee the absence of AEs. Also, the distortion of the latent geometry and its deviation from any Euclidean or ad-hoc structure is inevitable. Our current choice of StyleGAN-family generators is partly pragmatic: we optimize in a single, well defined latent space where it is tractable to define and analyse a metric induced from robust features and use the apparatus of differential geometry.
> > >
> > >   In latent diffusion models, there are multiple interacting spaces (the VAE latent, the time-dependent diffusion states, and conditioning latents), and the stochastic dynamics evolve over time. This makes it nontrivial to decide where to place a geometry, how to define geodesics consistently along the trajectory, and how a pullback metric should interact with the score dynamics.  In the revised version, we explicitly note that extending an induced geometry to the latent spaces and trajectories of diffusion models is an interesting and nontrivial direction for future work, and that a careful treatment would be required to study adversarial behaviour in that setting.
> > >
> > > **On Minors:**
> > > Thanks for pointing out these issues. We have corrected the typos in the revised manuscript and adjusted the corresponding sentences for clarity.
> > >
> > > Regarding the SoftPlus remark: all reported experiments use ReLU. SoftPlus was mentioned as a theoretically motivated alternative to ensure smoothness in the latent-to-ambient map, enabling a Riemannian manifold structure when activations are smooth and monotone. In practice, we found no measurable difference when swapping ReLU for SoftPlus, so the choice has no operational impact on results. We note that SoftPlus remains a valid option in settings requiring strict smoothness or alternative architectures.
> > >
> > > In the revised manuscript, we have now made the two-phase optimization details more clear in the main text.
> > >
> > > [1] Debenedetti et al. *A Light Recipe to Train Robust Vision Transformers*

---

> > > > ### Comment · Reviewer_RJrE · 2025-11-21
> > > >
> > > > Thank you for addressing my questions. I know I raised quite a few points, and I appreciate the significant effort you put into your response.
> > > >
> > > > I have one final request before updating my score. I fully agree that PCG aims to go beyond 'just flipping the label,' and I value this direction—I believe it is necessary for the domain of VCEs to advance. I also understand why the success rate was not reported; it might make PCG appear numerically inferior to baseline methods that prioritize pure efficiency over geometric properties.
> > > >
> > > > However, I argue that these numbers *must* be included in the paper for completeness. Without them, we risk a scenario where the method's actual effectiveness is opaque. Even if theoretically valid, a method could fail to flip the label entirely, which would indicate a failure to solve the optimization problem. I am well aware of the pressure in current publishing to produce 'tables with bold values,' but I strongly disagree with that culture. I value transparency over SOTA numbers in this context.

---

> ### Author Response · Authors · 2025-11-21
>
> Thank you again for your very thoughtful engagement with the paper and for emphasizing the importance of transparency. We fully agree that reporting a flip rate is important for completeness and to rule out degenerate cases where optimization fails to change the prediction at all.
>
> In the revised manuscript, we now report flip rates in **Table 2** in the main text for completeness, and we additionally discuss them in **Appendix B.6**. For the baseline methods, this metric has the usual interpretation: optimization starts from the source image, and the flip rate measures how often the procedure reaches a counterfactual that actually changes the classifier’s prediction to the target class.
>
> For PCG, the setup is different by construction: we fix the input and initialize a path to an endpoint that is already in the target class, then “pull” that geodesic back towards the input under the robust metric while trying to retain the target label. In this setting, the same number is better read as a target-retention rate under geometric constraints (how often the endpoint stays in the target region as we shorten the path). We adopt the standard “flip rate” terminology for consistency across methods, but explicitly highlight this distinction in **Appendix B.6** so the reader can interpret the numbers appropriately. Empirically, all methods—including PCG and the baselines—achieve high and broadly comparable flip/retention rates, which reinforces our earlier point that this metric alone cannot distinguish robust, semantically meaningful counterfactuals from adversarial solutions and must be read alongside the geometric and robustness-aware evaluations.

---

> > ### Comment · Reviewer_RJrE · 2025-11-23
> >
> > Thanks for that, I appreciate the addition and consider it an important part of the final version of the paper. I'm increasing my score to 6 and will be watching closely the further discussion among the other reviewers.

---

> > > ### Author Response · Authors · 2025-11-27
> > >
> > > Thank you for your thoughtful feedback throughout the process. We appreciate your reconsideration of the score.

---

### Official Review · Reviewer_Bs9x · 2025-11-03

**Soundness:** 3
**Presentation:** 3
**Contribution:** 3
**Rating:** 8
**Confidence:** 4

**Summary:**

This paper presents a method, Perceptual Counterfactual Geodesics (PCG), for generating counterfactual explanations by optimizing trajectories in a generator's latent space. The core contribution is the introduction of a robust Riemannian metric to define the latent geometry. This is induced from the feature spaces of robust vision models. This perceptually-aligned geometry enables the generation of smooth, semantically valid explanations that successfully avoid both off-manifold artifacts and on-manifold adversarial regions, which are common failure modes for existing latent-space methods.

**Strengths:**

- Strong motivation: Paper clearly articulates three speicific failure modes of prior latent space methods: 1. off-manifold traversal leading to artifacts, 2. local gradient optimization that ignores global structure, 3. generator exploitation of non-robust metrics.
- Perceptual Counterfactual Geodesics (PCG) is logical and well-structured. It employs a two-phase optimization process. This involves first finding an energy-minimizing geodesic, and tehn jointly refining the path and its endpoint with a classification loss.
- The experimental validation is comprehensive. Uses three high dimensional datasets (AFHQ, FFHQ, Plant Village), and two different generator architectures (STYLEGAN2, STYLEGAN3).
- The paper provides compelling visual evidence that PCG produces smooth, semantically coherent explainations while baselines collapse into off-manifold artifacts or on-manifold AEs. PCG also acheives the best scores in all geometry aware metrics.
- The authors introduce a robust, geometry-aware evaluation metric, $ L_R $. This proves to be more faithful to perceptual similarity and successfully exposes teh adversarial failure modes of baselines that remain hidden under standard $ L_1 $, $ L_2 $, or even non-robust feature metrics, $ L_F $.

**Weaknesses:**

- The paper's own conclusion frames it's contribution as operationalizing established ideas from pullback geometry and robust perception. The technical novelty is incremental, rather than a fundamental new theory.
- The paper's main quantitative results is incomplete, with runtime comparisions relegated to appendix. There is limited discussion of scalability to longer paths or higher resolution images.
- The justification for why robust features necessarily yield semantically meaningful geodesics rests on empirical demonstrations and appeals to prior adversarial robustness literature. It could be strengthened with a self contained theoretical proof for why this alignment must exist.
- The paper notes that the baseline RSGD methods were mostly proposed for tabular data settings. The authors had to implement their own adaptations for the vision domain, which could mean the comparison isn't fully representative of those methods intended use, and hence somewhat unfair.
- The sensitivity analysis in Appendix B.3 is limited. It relies on only three initialization runs per input and uses a single perceptual metric (LPIPS) for comparison. A broader statistical analysis would make the claims of low sensitivity more robust.

**Questions:**

(restating some of the weaknesses)
1. Can you provide theoretical analysis or intuition for why (and when) robust features guarantee semantically meaningful geodesics? Are there failure cases where robustness alone is insufficient?
2. The runtime analysis in Appendix B.5 mentions using T=10 nodes for the path. How does the computational cost (both time and memory) scale as T increases? Is there a practical limit. What is the trade-off between a smoother geodesic and computational cost?

---

> ### Author Response · Authors · 2025-11-21
>
> Responses to Weaknesses \& Questions
> 1. **Contribution**: While we build on established ideas from pullback geometry and robust perception, our work reframes the counterfactual optimization problem itself. We introduce a principled, novel two-stage optimization framework grounded in robust Riemannian geometry. This formulation unifies several previously disconnected ideas from robust representation learning, manifold-conforming generation, and geometric path optimization into a single framework that directly addresses longstanding failure modes (e.g., adversarial collapse, semantic drift, off-manifold traversal). By inducing geometry from robust feature spaces, our framework redefines counterfactual generation as a geometric process grounded in the representational structure of robust models, producing counterfactuals that remain intrinsically tethered to both the data manifold and robust semantics, in a way that prior methods show strong failure modes. In this sense, our contribution can be viewed a conceptual and technical shift in how counterfactual optimization is framed and grounded.
>
>
> 2. **On Scalability and Sensitivity Analysis**:  We thank the reviewers for raising these points. We have now conducted an extended runtime, smoothness, and sensitivity analysis to more thoroughly characterize the computational and stability properties of our method.
>
>      - **(Runtime and path-length scaling)**. To evaluate scalability with respect to the number of path nodes, we varied the path length $T \in \{10, 15, 20, 25\}$ and measured wall-clock time and peak CUDA memory for $512 \times 512$ images on a single GPU. As expected from the structure of our two-stage optimization, runtime grows approximately linearly in $T$, while memory increases more gently. For example, increasing $T$ from 10 to 25 raises the per-sample runtime from 3.4 to 10.9 minutes and peak memory from 23.4 GB to 27.3 GB. We also compare against RSGD/RSGD-C and show that, at fixed resolution and $T{=}10$, PCG is faster and uses less memory than these natural-gradient baselines despite optimizing a full path. These results are now reported in **Appendix B.5**.
>
>    - **(Smoothness–compute trade-off)**. There is a natural trade-off between geodesic smoothness and computational cost. We quantify this by measuring path-based average LPIPS and robust LPIPS step sizes, $\overline{\Delta \mathrm{LPIPS}}$ and $\overline{\Delta \mathrm{R\mbox{-}LPIPS}}$, for different path lengths $T \in \{3, 5, 8, 10, 15, 20\}$. Very short paths ($T{=}3, 5$) produce noticeably larger perceptual jumps between successive nodes, while increasing to $T=8\text{–}10$ yields a substantial reduction in both metrics.  Beyond $T{=}10$, improvements become marginal, with $T{=}15$ and $T{=}20$ offering only small smoothness gains at a near-linear increase in runtime and a moderate increase in peak memory. In practice, we therefore recommend $T \approx 8\text{–}10$ as a practical range. The corresponding trends are provided in **Figure 11** and discussed in **Appendix B.5**.
>
>     - **(Sensitivity to initialization and robustness.)**. To address concerns about sensitivity, we extended the analysis in **Appendix B.3** to use $M{=}15$ independent runs per input with different target-class initializations. We then evaluate the dispersion of the resulting counterfactual endpoints using LPIPS and its robust counterpart R-LPIPS.
>
>       We report:
>           (i) a **CF dispersion ratio** that compares intra-counterfactual variation to typical target-class variation,
>          (ii) an **initialization–output contraction ratio** that measures how much the optimization contracts diversity across different target exemplars, and
>         (iii) the **maximum pairwise distance** (“CF diameter”) among endpoints.
>
>         Across tasks, PCG counterfactuals form clusters that are significantly tighter than generic target-class variability, and the optimization consistently contracts diverse initializations into a small, stable neighborhood around the input in both standard and robust perceptual spaces. These results quantitatively support our claim that PCG converges to perceptually stable counterfactuals under different seeds and target exemplars (see **Appendix B.5**).

---

> ### Author Response · Authors · 2025-11-21
>
> 2. Continued
> **Image resolution and scalability.** We also assess scalability across resolutions by running PCG at $256^2$, $512^2$ (our default), and $1024^2$ using StyleGAN2 generators trained at each scale. With $T{=}10$, runtime scales roughly with the number of pixels (from 1.7 to 3.4 to 7.8 minutes per sample), while peak CUDA memory grows more moderately (from 19.8 GB to 27.7 GB).
>   Since all optimization is performed in latent space, the dominant cost stems from generator and robust-backbone forward/backward passes rather than pixel-wise operations, and we observe no qualitative change in convergence behavior or path quality across these resolutions. In particular, PCG remains stable and tractable up to $1024^2$ on a single 24–40 GB GPU.  Full resolution-scaling results are summarized in **Appendix B.5**.
>
> 3. **Adaptation of RSGD to the vision domain.** We appreciate the reviewer’s concern regarding the adaptation of RSGD to our vision setting. While RSGD was originally demonstrated in tabular settings, the method itself is not restricted to low-dimensional domains; rather, its practical deployment outside tabular data has been limited largely because the stochastic component of its pullback metric is computationally intensive even in low dimensions.
>
>    We adopt the deterministic part of the RSGD metric, consistent with deterministic generators such as StyleGAN2/3. Importantly, the same adaptation is applied symmetrically to both our method and the RSGD baselines, ensuring that the comparison is conducted under the same modeling assumptions and computational constraints.

---

> > ### Author Response · Authors · 2025-11-21
> >
> > **Justifications of the robust metric.** We thank the reviewer for raising this point, and we would like to add more justification and grounding for our proposed metric, and how, under well‑studied and empirically verified conditions, robust representations induce a geometry in which geodesic paths are strongly biased toward semantically meaningful directions along the data manifold.
> >
> > - **Grounding.** First, adversarial training has been shown to change the relationship between gradients and the data manifold. [1] quantify that robust models exhibit saliency maps more strongly aligned with the input structure and suppress high-frequency, non-salient directions, providing a concrete link between robustness and interpretable gradients. [2] then give a theoretical explanation via *off-manifold robustness*: if a model is trained to be more robust off the data manifold than on it, its input gradients are forced to lie approximately in the tangent bundle of the manifold, which in turn explains why robust models tend to have *Perceptually Aligned Gradients* (PAGs) [3, 4]. These results make precise that robustness reshapes the local geometry so that gradient-based changes follow intrinsic manifold directions rather than adversarial “spikes” orthogonal to the data.
> >
> >  - **Human perceptual alignment via robustness.** Second, this manifold alignment translates directly into improved perceptual alignment. The literature on PAGs and robust perceptual metrics shows that distances in robust feature spaces correlate better with human similarity judgments than in standard ones. [5] demonstrate that LPIPS-type metrics are vulnerable and can be rectified to become robust to adversarial perturbations. [6] go further and train a 1-Lipschitz, provably robust perceptual similarity metric, showing that robustness in the feature space leads simultaneously to increased adversarial stability and better alignment with human perceptual similarity. Together with the analysis of [2], these works support the view that infinitesimal moves in robust feature space correspond to smooth, semantically coherent deformations along the data manifold.
> >
> >  - **Our metric.** Our robust pullback metric is designed to encode exactly this structure. By aggregating features from a robust backbone and pulling them back through the generator, we define a Riemannian geometry that suppresses high-frequency, or adversarial changes. Geodesics under this metric are therefore biased to track the manifold-aligned, perceptually meaningful directions that robust models use internally, rather than the brittle directions that standard Euclidean pixel space or fragile feature spaces often follow.
> >
> > - **When robustness alone is insufficient.** At the same time, we do not present this as a universal guarantee. The induced geometry can degrade when there is a strong distribution shift between the robust model’s training domain and the generator’s domain. Similarly, if the generator does not faithfully parameterize the data manifold (e.g., known failure modes in GANs such as mode collapse; [7]), no metric can fully prevent semantically problematic traversals. Our induced geometry is most trustworthy when (i) the generator and robust backbone operate on approximately the same domain, and (ii) geodesic paths are reasonably well sampled with enough discretization of the underlying robust geodesic to allow the optimizer to distribute smooth semantic change across more intermediate states (see **Appendix B.5** on how the number of points on the path can affect smoothness).
> >
> >    Under strong distribution shifts, under-sampled paths, or strong displacements along trajectories that leave these neighborhoods, the semantic interpretation of the metric can degrade even though the construction remains mathematically valid. We have now included this discussion in the revision.
> >
> > [1] Etmann et al. *On the Connection Between Adversarial Robustness and Saliency Map Interpretability*
> > [2] Srinivas et al. *Which Models have Perceptually‑Aligned Gradients? An Explanation via Off‑Manifold Robustness*
> > [3] Ganz et al. *Do Perceptually Aligned Gradients Imply Adversarial Robustness?*
> > [4] Kaur et al. *Are Perceptually-Aligned Gradients a General Property of Robust Classifiers?*
> > [5] Kettunen et al. *E-LPIPS: Robust Perceptual Image Similarity via Random Transformation Ensembles*
> > [6] Ghazanfari et al. *LipSim: A Provably Robust Perceptual Similarity Metric*
> > [7] Karras et al. *Analyzing and Improving the Image Quality of StyleGAN*

---

> ### Comment · Reviewer_Bs9x · 2025-11-26
> **Maintain acceptance recommendation**
>
> I appreciate the comprehensive response addressing my comments on weakness and questions. The extended analysis looks thorough, and the 10-run sensitivity analysis with quantitative metrics demonstrates stability.
>
> I maintain my acceptance recommendation.

---

> > ### Author Response · Authors · 2025-11-27
> >
> > Thank you for your follow-up. We appreciate your engagement and are glad the rebuttal addressed your concerns.

---

### Author Response · Authors · 2025-11-21

We thank all reviewers for their thoughtful feedback. Below, we respond to each review separately. A recurring suggestion was to strengthen the quantitative analysis; we have done so in the revised manuscript, with all new results and extensions highlighted in red.

---

### Comment · Area_Chair_viox · 2025-11-23

Dear Reviewers,

The authors have submitted their rebuttal addressing your reviews. Please take the time to:

1. Read the rebuttal carefully
2. Ask clarifying questions if anything remains unclear
3. Update your scores and reviews based on the authors' responses

Please be mindful of timing: If you have follow-up questions for the authors, **post them early enough to give them adequate time to respond** before the discussion period closes on December 3rd.

Your timely engagement is crucial for a fair and thorough review process.

Thank you for your continued effort on this paper.

Best regards,
Area Chair

---

### Author Response · Authors · 2025-11-30
**Authors’ final comment to Reviewers, Area Chairs, and Program Chairs**

Dear Reviewers, Area Chair, and Program Chairs,

We first want to express our understanding of, and respect for, the ICLR program chairs’ decision to revert reviews to their pre-discussion state and to reassign papers to new area chairs in response to the recent OpenReview incident. We appreciate the clear communication around these measures and recognize that they are motivated by a sincere commitment to the integrity and fairness of the review process.

For our submission, we would like to provide some context for the new area chair and for the record. As visible in the discussion thread, the reviewers engaged with us in a timely, detailed, and technically deep manner throughout the rebuttal period. In response to their comments, we carried out a substantial number of additional experiments and clarifications and iteratively revised the manuscript. **Before the public disclosure of the OpenReview bug on November 27, the discussion for our paper had effectively concluded:** all three reviewers explicitly stated in their final comments that their main concerns had been addressed and that they had maintained or increased their scores. In particular, as can be seen from their final responses and the accompanying score changes in the discussion below, **Reviewer Bs9x** maintained their acceptance recommendation (**score 8**), **Reviewer RJrE** increased their score from **4 to 6**, and **Reviewer T7oH** increased their score from **4 to 8**, taking the **overall scores from (8, 4, 4) to (8, 6, 8)**. Our latest revision reflects all changes and additional experiments described during this rebuttal.

We fully understand that, under the new policy, only the original “Official Review” text and pre-discussion scores are now displayed, and that reviewers can no longer modify scores or continue the conversation. Our sole aim with this comment is to ensure that the area chair and program chairs have a clear perspective on the trajectory of the discussion: the detailed back-and-forth, the additional experiments and analyses we carried out, and the reviewers’ own written statements and score updates indicating that their impressions of the work improved on this basis. We trust that this full context will be taken into account when forming the meta-review and final recommendation.

We are genuinely grateful to the reviewers for the seriousness and constructiveness of their engagement. Their comments and follow-up discussion significantly improved the paper, and we deeply appreciate the time and care they invested, especially under difficult circumstances for the broader community.

Thank you to the reviewers, area chairs, and program chairs for your efforts in safeguarding both the quality and fairness of the conference.

Regards,
Authors

---

### Meta-Review · Area_Chair_Mm5s · 2026-01-05

**Summary:**

The reviewers acknowledged the motivation behind this work and the proposed algorithm, but expressed some concerns related to the empirical evaluation, such as scalability, statistical significance, sensitivity to hyper-parameters, robustness of metrics, datasets used, and comparison to the state of the art.

**Reviewer Concerns:**

The authors convincingly addressed the reviewer's concerns, with maybe an exception related to the datasets used (w.r.t. existing methods).

**Reviewer Scores:**

As a matter of fact, there was a discussion prior to the security breach, and the discussion between the reviewers shows that two reviewers were willing to raise their scores, from 4 to 6 and from 4 to 8, respectively. There is therefore a clear consensus for acceptance.

---

### Decision · Program_Chairs · 2026-01-26

Accept (Poster)